# Low-rank Optimal Transport:
# Approximation, Statistics and Debiasing

**Meyer Scetbon**
CREST, ENSAE
meyer.scetbon@ensae.fr

**Marco Cuturi**
Apple and CREST, ENSAE
cuturi@apple.com

## Abstract

The matching principles behind optimal transport (OT) play an increasingly important role in machine learning, a trend which can be observed when OT is used to disambiguate datasets in applications (e.g. single-cell genomics) or used to improve more complex methods (e.g. balanced attention in transformers or self-supervised learning). To scale to more challenging problems, there is a growing consensus that OT requires solvers that can operate on millions, not thousands, of points. The low-rank optimal transport (LOT) approach advocated in Scetbon et al. [2021] holds several promises in that regard, and was shown to complement more established entropic regularization approaches, being able to insert itself in more complex pipelines, such as quadratic OT. LOT restricts the search for low-cost couplings to those that have a low-nonnegative rank, yielding linear time algorithms in cases of interest. However, these promises can only be fulfilled if the LOT approach is seen as a legitimate contender to entropic regularization when compared on properties of interest, where the scorecard typically includes theoretical properties (statistical complexity and relation to other methods) or practical aspects (debiasing, hyperparameter tuning, initialization). We target each of these areas in this paper in order to cement the impact of low-rank approaches in computational OT.

## 1 Introduction

Optimal transport (OT) is used across data-science to put in correspondence different sets of observations. These observations may come directly from datasets, or, in more advanced applications, depict intermediate layered representations of data. OT theory provides a single grammar to describe and solve increasingly complex matching problems (linear, quadratic, regularized, unbalanced, etc...), making it gain a stake in various areas of science such as as single-cell biology Schiebinger et al. [2019], Yang et al. [2020], Demetci et al. [2020], imaging Schmitz et al. [2018], Heitz et al. [2020], Zheng et al. [2020] or neuroscience Janati et al. [2020], Koundal et al. [2020].

**Regularized approaches to OT.** Solving OT problems at scale poses, however, formidable challenges. The most obvious among them is computational: the Kantorovich [1942] problem on discrete measures of size $n$ is a linear program that requires $O(n^3 \log n)$ operations to be solved. A second and equally important challenge lies in the estimation of OT in high-dimensional settings, since it suffers from the curse-of-dimensionality Fournier and Guillin [2015]. The advent of regularized approaches, such as entropic regularization [Cuturi, 2013], has pushed these boundaries thanks for faster algorithms [Scetbon and Cuturi, 2020, Chizat et al., 2020, Clason et al., 2021] and improved statistical aspects [Genevay et al., 2018a]. Despite these clear strengths, regularized OT solvers remain, however, costly as they typically scale quadratically in the number of observations.

**Scaling up OT using low-rank couplings.** While it is always intuitively possible to reduce the size of measures (e.g. using $k$-means) prior to solving an OT between them, a promising line of work proposes to combine both [Forrow et al., 2019, Scetbon et al., 2021, 2022]. Conceptually, these

low-rank approaches solve simultaneously both an optimal clustering/aggregation strategy with the computation of an effective transport. This intuition rests on an explicit factorization of couplings into two sub-couplings. This has several computational benefits, since its computational cost becomes linear in $n$ if the ground cost matrix seeded to the OT problem has itself a low-rank. While these computational improvements, mostly demonstrated empirically, hold several promises, the theoretical properties of these methods are not yet well established. This stands in stark contrast to the Sinkhorn approach, which is comparatively much better understood.

**Our Contributions.** The goal of this paper is to advance our knowledge, understanding and practical ability to leverage low-rank factorizations in OT. This paper provides five contributions, targeting theoretical and practical properties of LOT: *(i)* We derive the rate of convergence of the low-rank OT to the true OT with respect to the non-nnegative rank parameter. *(ii)* We make a first step towards a better understanding of the statistical complexity of LOT by providing an upper-bound of the statistical error, made when estimating LOT using the plug-in estimator; that upper-bound has a parametric rate $\mathcal{O}(\sqrt{1/n})$ that is independent of the dimension. *(iii)* We introduce a debiased version of LOT: as the Sinkhorn divergence [Feydy et al., 2018], we show that debiased LOT is nonnegative, metrizes the weak convergence, and that it interpolates between the maximum mean discrepancy [Gretton et al., 2012] and OT. *(iv)* We exhibit links between the bias induced by the low-rank factorization and clustering methods. *(v)* We propose practical strategies to tune the step-length and the initialization of the algorithm in [Scetbon et al., 2021].

**Notations.** We consider $(\mathcal{X}, d_{\mathcal{X}})$ and $(\mathcal{Y}, d_{\mathcal{Y}})$ two nonempty compact Polish spaces and we denote $\mathcal{M}_1^+(\mathcal{X})$ (resp. $\mathcal{M}_1^+(\mathcal{Y})$) the space of positive Radon probability measures on $\mathcal{X}$ (resp. $\mathcal{Y}$). For all $n \geq 1$, we denote $\Delta_n$ the probability simplex of size $n$ and $\Delta_n^*$ the subset of $\Delta_n$ of positive histograms. We write $\mathbf{1}_n \triangleq (1, \ldots, 1)^T \in \mathbb{R}^n$ and we denote similarly $\| \cdot \|_2$ the Euclidean norm and the Euclidean distance induced by this norm depending on the context.

## 2  Background on Low-rank Optimal Transport

Let $\mu \in \mathcal{M}_1^+(\mathcal{X})$, $\nu \in \mathcal{M}_1^+(\mathcal{Y})$ and $c : \mathcal{X} \times \mathcal{Y} \to \mathbb{R}_+$ a nonnegative and continuous function. The Kantorovitch formulation of optimal transport between $\mu$ and $\nu$ is defined by

$$\text{OT}_c(\mu, \nu) \triangleq \min_{\pi \in \Pi(\mu, \nu)} \int_{\mathcal{X} \times \mathcal{Y}} c(x, y) d\pi(x, y), \tag{1}$$

where the feasible set is the set of distributions over the product space $\mathcal{X} \times \mathcal{Y}$ with marginals $\mu$ and $\nu$:

$$\Pi(\mu, \nu) \triangleq \left\{ \pi \in \mathcal{M}_1^+(\mathcal{X} \times \mathcal{Y}) \text{ s.t. } P_{1\#\pi} = \mu, P_{2\#\pi} = \nu \right\},$$

with $P_{1\#\pi}$ (resp. $P_{2\#\pi}$), the pushforward probability measure of $\pi$ using the projection maps $P_1(x, y) = x$ (resp. $P_2(x, y) = y$). When there exists an optimal coupling solution of (1) supported on a graph of a function, we call such function a Monge map. In the discrete setting, one can reformulate the optimal transport problem as a linear program over the space of nonnegative matrices satisfying the marginal constraints. More precisely, let $a$ and $b$ be respectively elements of $\Delta_n^*$ and $\Delta_m^*$ and let also $\mathbf{X} \triangleq \{x_1, \ldots, x_n\}$ and $\mathbf{Y} \triangleq \{y_1, \ldots, y_m\}$ be respectively two subsets of $\mathcal{X}$ and $\mathcal{Y}$. By denoting $\mu_{a,\mathbf{X}} \triangleq \sum_{i=1}^n a_i \delta_{x_i}$ and $\nu_{b,\mathbf{Y}} \triangleq \sum_{j=1}^m b_j \delta_{y_j}$ the two discrete distributions associated and writing $C \triangleq [c(x_i, y_j)]_{i,j}$, the discrete optimal transport problem can be formulated as

$$\text{OT}_c(\mu_{a,\mathbf{X}}, \nu_{b,\mathbf{Y}}) = \min_{P \in \Pi_{a,b}} \langle C, P \rangle \text{ where } \Pi_{a,b} \triangleq \{P \in \mathbb{R}_+^{n \times m} \text{ s.t. } P\mathbf{1}_m = a, P^T\mathbf{1}_n = b\}. \tag{2}$$

[Scetbon et al. [2021]] propose to constrain the discrete optimal transport problem to couplings that have a low-nonnegative rank:

**Definition 1.** *Given $M \in \mathbb{R}_+^{n \times m}$, the nonnegative rank of $M$ is defined by:* $\text{rk}_+(M) \triangleq \min\{q | M = \sum_{i=1}^q R_i, \forall i, \text{rk}(R_i) = 1, R_i \geq 0\}$.

Note that for any $M \in \mathbb{R}_+^{n \times m}$, we always have that $\text{rk}_+(M) \leq \min(n, m)$. For $r \geq 1$, we consider the set of couplings satisfying marginal constaints with nonnegative-rank of at most $r$ as $\Pi_{a,b}(r) \triangleq \{P \in \Pi_{a,b}, \text{rk}_+(P) \leq r\}$. The discrete Low-rank Optimal Transport (LOT) problem is defined by:

$$\text{LOT}_{r,c}(\mu_{a,\mathbf{X}}, \nu_{b,\mathbf{Y}}) \triangleq \min_{P \in \Pi_{a,b}(r)} \langle C, P \rangle. \tag{3}$$

To solve this problem, Scetbon et al. [2021] show that Problem (3) is equivalent to

$$\min_{(Q,R,g)\in\mathcal{C}_1(a,b,r)\cap\mathcal{C}_2(r)}\langle C, Q\operatorname{diag}(1/g)R^T\rangle,\tag{4}$$

where $\mathcal{C}_1(a,b,r)\triangleq\left\{(Q,R,g)\in\mathbb{R}_+^{n\times r}\times\mathbb{R}_+^{m\times r}\times(\mathbb{R}_+^*)^r\text{ s.t. }Q\mathbf{1}_r=a, R\mathbf{1}_r=b\right\}$ and $\mathcal{C}_2(r)\triangleq\left\{(Q,R,g)\in\mathbb{R}_+^{n\times r}\times\mathbb{R}_+^{m\times r}\times\mathbb{R}_+^r\text{ s.t. }Q^T\mathbf{1}_n=R^T\mathbf{1}_m=g\right\}$. They propose to solve it using a mirror descent scheme and prove the non-asymptotic stationary convergence of their algorithm. While Scetbon et al. [2021] only focus on the discrete setting, we consider here its extension for arbitrary probability measures. Following [Forrow et al., 2019], we define the set of rank-$r$ couplings satisfying marginal constraints by:

$$\Pi_r(\mu,\nu)\triangleq\{\pi\in\Pi(\mu,\nu):\exists(\mu_i)_{i=1}^r\in\mathcal{M}_1^+(\mathcal{X})^r, (\nu_i)_{i=1}^r\in\mathcal{M}_1^+(\mathcal{Y})^r, \lambda\in\Delta_r^*\text{ s.t. }\pi=\sum_{i=1}^r\lambda_i\mu_i\otimes\nu_i\}.$$

This more general definition of LOT between $\mu\in\mathcal{M}_1^+(\mathcal{X})$ and $\nu\in\mathcal{M}_1^+(\mathcal{Y})$ reads:

$$\operatorname{LOT}_{r,c}(\mu,\nu)\triangleq\inf_{\pi\in\Pi_r(\mu,\nu)}\int_{\mathcal{X}\times\mathcal{Y}}c(x,y)d\pi(x,y).\tag{5}$$

Note that this definition of $\operatorname{LOT}_{r,c}$ is consistent as it coincides with the one defined in (3) on discrete probability measures. Observe also that $\Pi_r(\mu,\nu)$ is compact for the weak topology and therefore the infimum in (5) is attained. See Appendix A for more details.

## 3 Approximation Error of LOT to original OT as a function of rank

Our goal in this section is to obtain a control of the error induced by the low-rank constraint when trying to approximate the true OT cost. We provide first a control of the approximation error in the discrete setting. The proof is given in Appendix B.1.

**Proposition 1.** *Let $n,m\geq 2$, $\mathbf{X}\triangleq\{x_1,\dots,x_n\}\subset\mathcal{X}$, $\mathbf{Y}\triangleq\{y_1,\dots,y_m\}\subset\mathcal{Y}$ and $a\in\Delta_n^*$ and $b\in\Delta_m^*$. Then for $2\leq r\leq\min(n,m)$, we have that*

$$|\operatorname{LOT}_{r,c}(\mu_{a,\mathbf{X}},\nu_{b,\mathbf{Y}})-\operatorname{OT}_c(\mu_{a,\mathbf{X}},\nu_{b,\mathbf{Y}})|\leq\|C\|_\infty\, ln(\min(n,m)/(r-1))$$

**Remark 1.** *Note that this result improves the control obtained in [Liu et al., 2021], where they obtain that $|\operatorname{LOT}_{r,c}(\mu_{a,\mathbf{X}},\nu_{b,\mathbf{Y}})-\operatorname{OT}_c(\mu_{a,\mathbf{X}},\nu_{b,\mathbf{Y}})|\lesssim\|C\|_\infty\sqrt{nm}(\min(n,m)-r)$ as we have for any $z,z'\geq 1$, $|\ln(z)-\ln(z')|\leq|z-z'|$.*

It is in fact possible to obtain another control of the approximation error by partitioning the space where the measures are supported. For that purpose let us introduce the notion of entropy numbers.

**Definition 2.** *Let $(\mathcal{Z},d)$ a metric space, $\mathcal{W}\subset\mathcal{Z}$ and $k\geq 1$ an integer. Then by denoting $B_\mathcal{Z}(z,\varepsilon)\triangleq\{y\in\mathcal{Z}:\ d(z,y)\leq\varepsilon\}$, we define the $k$-th (dyadic) entropy number of $\mathcal{W}$ as*

$$\mathcal{N}_k(\mathcal{W},d)\triangleq\inf\{\varepsilon\text{ s.t. }\exists z_1,\dots,z_{2^k}\in\mathcal{Z}:\ \mathcal{W}\subset\cup_{i=1}^{2^k}B_\mathcal{Z}(z_i,\varepsilon)\}.$$

For example, any compact set $\mathcal{W}$ of $\mathbb{R}^d$ admits finite entropy numbers, and by denoting $R\triangleq\sup_{w\in\mathcal{W}}\|w\|_2$, we have $\mathcal{N}_k(\mathcal{W},\|\cdot\|_2)\leq 4R/2^{k/d}$. We obtain next a control of the approximation error of $\operatorname{LOT}_{r,c}$ to the true OT cost using entropy numbers (see proof in Appendix B.2).

**Proposition 2.** *Let $\mu\in\mathcal{M}_1^+(\mathcal{X})$, $\nu\in\mathcal{M}_1^+(\mathcal{Y})$ and assume that $c$ is $L$-Lipschitz w.r.t. $x$ and $y$. Then for any $r\geq 1$, we have*

$$|\operatorname{LOT}_{r,c}(\mu,\nu)-\operatorname{OT}_c(\mu,\nu)|\leq 2L\max(\mathcal{N}_{\lfloor\log_2(\lfloor\sqrt{r}\rfloor)\rfloor}(\mathcal{X},d_\mathcal{X}),\mathcal{N}_{\lfloor\log_2(\lfloor\sqrt{r}\rfloor)\rfloor}(\mathcal{Y},d_\mathcal{Y}))$$

This results in the following bound for the $p$-Wasserstein distance for any $p\geq 1$ on $\mathbb{R}^d$.

**Corollary 1.** *Let $d\geq 1$, $p\geq 1$, $\mathcal{X}$ a compact subspace of $\mathbb{R}^d$ and $\mu,\nu\in\mathcal{M}_1^+(\mathcal{X})$. By denoting $R\triangleq\sup_{x\in\mathcal{X}}\|x\|_2$, we obtain that for any $r\geq 1$,*

$$|\operatorname{LOT}_{r,\|\cdot\|_2^p}(\mu,\nu)-\operatorname{OT}_{\|\cdot\|_2^p}(\mu,\nu)|\leq 4dp\frac{(8R^2)^p}{r^{p/2d}}.$$

As per the Proof of Proposition 2 we can provide a tighter control, assuming a Monge map exists.

**Corollary 2.** *Under the same assumptions of Proposition 2 and by assuming in addition that there exists a Monge map solving $OT_c(\mu, \nu)$, we obtain that for any $r \geq 1$,*

$$|\text{LOT}_{r,c}(\mu, \nu) - \text{OT}_c(\mu, \nu)| \leq L\mathcal{N}_{\lfloor \log_2(r) \rfloor}(\mathcal{Y}, d_{\mathcal{Y}}).$$

When $\mathcal{X} = \mathcal{Y}$ are a subspaces of $\mathbb{R}^d$, a sufficient condition for a Monge map to exists is that either $\mu$ or $\nu$ is absolutely continuous with respect to the Lebesgue measure and that $c$ is of the form $h(x - y)$ where $h : \mathcal{X} \to \mathbb{R}_+$ is a strictly convex function [Santambrogio, 2015, Theorem 1.17]. Therefore if $\mu$ is absolutely continuous with respect to the Lebesgue measure, we obtain for any $r \geq 1$ and $p > 1$

$$|\text{LOT}_{r,\|\cdot\|_2^p}(\mu, \nu) - \text{OT}_{\|\cdot\|_2^p}(\mu, \nu)| \leq 2dp\frac{(8R^2)^p}{r^{p/d}}.$$

# 4 Sample Complexity of LOT

We now focus on the statistical performance of the plug-in estimator for LOT. In the following we assume that $\mathcal{X} = \mathcal{Y}$ for simplicity. Given $\mu, \nu \in \mathcal{M}_1^+(\mathcal{X})$, we denote the empirical measures associated $\hat{\mu}_n \triangleq \frac{1}{n}\sum_{i=1}^n \delta_{X_i}$ and $\hat{\nu}_n \triangleq \frac{1}{n}\sum_{i=1}^n \delta_{Y_i}$, where $(X_i, Y_i)_{i=1}^n$ are sampled independently from $\mu \otimes \nu$. We consider the plug-in estimator defined as $\text{LOT}_{r,c}(\hat{\mu}_n, \hat{\nu}_n)$, and we aim at quantifying the rate at which it converges towards the true low-rank optimal transport cost $\text{LOT}_{r,c}(\mu, \nu)$. Before doing so, in the next Proposition we show that this estimator is consistent on compact spaces. The proof is given in Appendix B.3.

**Proposition 3.** *Let $r \geq 1$ and $\mu, \nu \in \mathcal{M}_1^+(\mathcal{X})$, then $\text{LOT}_{r,c}(\hat{\mu}_n, \hat{\nu}_n) \xrightarrow[n \to +\infty]{} \text{LOT}_{r,c}(\mu, \nu)$ a.s.*

Next we aim at obtaining the convergence rates of our plug-in estimator. In the following Proposition, we obtain a non-asymptotic upper-bound of the statistical error. See Appendix B.4 for the proof.

**Proposition 4.** *Let $r \geq 1$ and $\mu, \nu \in \mathcal{M}_1^+(\mathcal{X})$. Then, there exists a constant $K_r$ such that for any $\delta > 0$ and $n \geq 1$, we have, with a probability of at least $1 - 2\delta$, that*

$$\text{LOT}_{r,c}(\hat{\mu}_n, \hat{\nu}_n) \leq \text{LOT}_{r,c}(\mu, \nu) + 11\|c\|_\infty\sqrt{\frac{r}{n}} + K_r\|c\|_\infty\left[\sqrt{\frac{\log(40/\delta)}{n}} + \frac{\sqrt{r}\log(40/\delta)}{n}\right].$$

This result is, to the best of our knowledge, the first attempt at providing a statistical control of low-rank optimal transport. We provide an upper-bound of the plug-in estimator which converges towards $\text{LOT}_{r,c}$ at a parametric rate and which is independent of the dimension on general compact metric spaces. While we fall short of providing a lower bound that could match that upper bound, and therefore provide a complete statistical complexity result, we believe this result might provide a first explanation on why, in practice, $\text{LOT}_{r,c}$ displays better statistical properties than unregularized OT and its curse of dimensionality [Dudley, 1969]. In addition, that upper bound compares favorably to known results on entropic optimal transport. The rate of entropy regularized OT does not depend on the ambient dimension with respect to $n$, but carries an exponential dependence in dimension with respect to the regularization parameter $\varepsilon$ [Mena and Niles-Weed, 2019]. By contrast, the term associated with the nonnegative rank $r$ in our bound has no direct dependence on dimension.

Our next aim is to obtain an explicit rate with respect to $r$ and $n$. In Proposition 4, we cannot control explicitly $K_r$ in the general setting. Indeed, in our proof, we obtain that $K_r \triangleq 14/\min_i \lambda_i^*$ where $(\lambda_i^*)_{i=1}^r \in \Delta_r^*$ are the weights involved in the decomposition of one optimal solution of the true $\text{LOT}_{r,c}(\mu, \nu)$. Therefore the control of $K_r$ requires additional assumptions on the optimal solutions of $\text{LOT}_{r,c}(\mu, \nu)$. In the following Proposition, we obtain an explicit upper-bound of the plug-in estimator with respect to $r$ and $n$ in the asymptotic regime.

**Proposition 5.** *Let $r \geq 1$, $\delta > 0$ and $\mu, \nu \in \mathcal{M}_1^+(\mathcal{X})$. Then there exists a constant $N_{r,\delta}$ such that if $n \geq N_{r,\delta}$ then with a probability of at least $1 - 2\delta$, we have*

$$\text{LOT}_{r,c}(\hat{\mu}_n, \hat{\nu}_n) \leq \text{LOT}_{r,c}(\mu, \nu) + 11\|c\|_\infty\sqrt{\frac{r}{n}} + 77\|c\|_\infty\sqrt{\frac{\log(40/\delta)}{n}}.$$

Note that one cannot recover the result obtained in Proposition 5 from the one obtained in Proposition 4 as we have that $K_r \geq 14r \xrightarrow[r \to +\infty]{} +\infty$. In order to prove the above result, we use an extension of the McDiarmid's inequality when differences are bounded with high probability [Kutin, 2002]. See proof in Appendix B.5 for more details.

# 5    Debiased Formulation of LOT

We introduce here the debiased formulation of $\mathrm{LOT}_{r,c}$ and show that it is able to distinguish two distributions, metrize the convergence in law and can be used as a new objective in order to learn distributions. We focus next on the debiasing terms involving measures with themselves $\mathrm{LOT}_{r,c}(\mu, \mu)$ in this new divergence, and show that they can be interpreted as defining a new clustering method generalizing $k$-means for any geometry.

## 5.1    On the Proprieties of the Debiased Low-rank Optimal Transport

When it comes to learn (or generate) a distribution in ML applications given samples, it is crucial to consider a divergence that is able to distinguish between two distributions and metrize the convergence in law. In general, $\mathrm{LOT}_{r,c}(\mu, \mu) \neq 0$ and the minimum of $\mathrm{LOT}_{r,c}(\nu, \mu)$ with respect to $\nu$ will not necessarily recover $\mu$. In order to alleviate this issue we propose a debiased version of $\mathrm{LOT}_{r,c}$ defined for any $\mu, \nu \in \mathcal{M}_1^+(\mathcal{X})$ as

$$\mathrm{DLOT}_{r,c}(\mu, \nu) \triangleq \mathrm{LOT}_{r,c}(\mu, \nu) - \frac{1}{2}[\mathrm{LOT}_{r,c}(\mu, \mu) + \mathrm{LOT}_{r,c}(\nu, \nu)] \ .$$

Note that $\mathrm{DLOT}_{r,c}(\nu, \nu) = 0$. In the next Proposition, we show that, as the Sinkhorn divergence [Genevay et al., 2018b, Feydy et al., 2018], $\mathrm{DLOT}_{r,c}$ interpolates between the Maximum Mean Discrepancy (MMD) and OT. See proof in Appendix B.6.

**Proposition 6.** *Let $\mu, \nu \in \mathcal{M}_1^+(\mathcal{X})$. Let us assume that $c$ is symmetric, then we have*

$$\mathrm{DLOT}_{1,c}(\mu, \nu) = \frac{1}{2} \int_{\mathcal{X}^2} -c(x, y)d[\mu - \nu] \otimes d[\mu - \nu](x, y) \ .$$

*If in addition we assume the $c$ is Lipschitz w.r.t to $x$ and $y$, then we have*

$$\mathrm{DLOT}_{r,c}(\mu, \nu) \xrightarrow[r \to +\infty]{} \mathrm{OT}_c(\mu, \nu) \ .$$

Next, we aim at showing some useful properties of the debiased low-rank OT for machine learning applications. For that purpose, let us first recall some definitions.

**Definition 3.** *We say that the cost $c : \mathcal{X} \times \mathcal{X} \to \mathbb{R}_+$ is a semimetric on $\mathcal{X}$ if for all $x, x' \in \mathcal{X}$, $c(x, x') = c(x', x)$ and $c(x, x') = 0$ if and only if $x = x'$. In addition we say that $c$ has a negative type if $\forall n \geq 2$, $x_1, \ldots, x_n \in \mathcal{X}$ and $\alpha_1, \ldots, \alpha_n \in \mathbb{R}$ such that $\sum_{i=1}^n \alpha_i = 0$, $\sum_{i,j=1}^n \alpha_i \alpha_j c(x_i, x_j) \leq 0$. We say also that $c$ has a strong negative type if for all $\mu, \nu \in \mathcal{M}_1^+(\mathcal{X})$, $\mu \neq \nu \implies \int_{\mathcal{X}^2} c(x, y)d[\mu - \nu] \otimes [\mu - \nu] < 0$.*

Note that if $c$ has a strong negative type, then $c$ has a negative type too. For example, all Euclidean spaces and even separable Hilbert spaces endowed with the metric induced by their inner products have strong negative type. Also, on $\mathbb{R}^d$, the squared Euclidean distance has a negative type [Sejdinovic et al., 2013].

We can now provide stronger geometric guarantees for $\mathrm{DLOT}_{r,c}$. In the next Proposition, we show that for a large class of cost functions, $\mathrm{DLOT}_{r,c}$ is nonnegative, able to distinguish two distributions, and metrizes the convergence in law. The proof is given in Appendix B.8.

**Proposition 7.** *Let $r \geq 1$, and let us assume that $c$ is a semimetric of negative type. Then for all $\mu, \nu \in \mathcal{M}_1^+(\mathcal{X})$, we have that*

$$\mathrm{DLOT}_r(\mu, \nu) \geq 0 \ .$$

*In addition, if $c$ has strong negative type then we have also that*

$$\mathrm{DLOT}_{r,c}(\mu, \nu) = 0 \iff \mu = \nu \ \text{and}$$
$$\mu_n \to \mu \iff \mathrm{DLOT}_{r,c}(\mu_n, \mu) \to 0 \ .$$

*where the convergence of the sequence of probability measures considered is the convergence in law.*

Observe that when $c$ has strong negative type, $\nu \to \mathrm{DLOT}_{r,c}(\nu,\mu) \geq 0$ and it admits a unique global minimizer at $\nu = \mu$. Therefore, $\mathrm{DLOT}_{r,c}$ has desirable properties to be used as a loss. It is also worth noting that, in order to obtain the metrization of the convergence in law, we show the following Proposition. See proof in Appendix B.7.

**Proposition 8.** *Let $r \geq 1$ and $(\mu_n)_{n\geq 0}$ and $(\nu_n)_{n\geq 0}$ two sequences of probability measures such that $\mu_n \to \mu$ and $\nu_n \to \nu$ with respect to the convergence in law. Then we have that*

$$\mathrm{LOT}_{r,c}(\mu_n, \nu_n) \to \mathrm{LOT}_{r,c}(\mu, \nu) \ .$$

## 5.2 Low-Rank Transport Bias and Clustering

We turn next to the debiasing terms appearing in DLOT and exhibit links between LOT and clustering methods. Indeed, in the discrete setting, the low-rank bias of a probability measure $\mu$ defined as $\mathrm{LOT}_{k,c}(\mu,\mu)$ can be seen as a generalized version of the $k$-means method for any geometry. In the next Proposition we obtain a new formulation of $\mathrm{LOT}_{k,c}(\mu,\mu)$ viewed as a general clustering method on arbitrary metric space. See proof in Appendix B.9.

**Proposition 9.** *Let $n \geq k \geq 1$, $\mathbf{X} \triangleq \{x_1, \ldots, x_n\} \subset \mathcal{X}$ and $a \in \Delta_n^*$. If $c$ is a semimetric of negative type, then by denoting $C = (c(x_i, x_j))_{i,j}$, we have that*

$$\mathrm{LOT}_{k,c}(\mu_{a,\mathbf{X}}, \mu_{a,\mathbf{X}}) = \min_Q \langle C, Q diag(1/Q^T \mathbf{1}_n) Q^T \rangle \ \ s.t. \ Q \in \mathbb{R}_+^{n \times k} \ , \ Q\mathbf{1}_k = a \ . \quad (6)$$

Let us now explain in more details the link between (6) and $k$-means. When $\mathcal{X}$ is a subspace of $\mathbb{R}^d$, $c$ is the squared Euclidean distance and $a = \mathbf{1}_n$, we recover exactly the $k$-means algorithm.

**Corollary 3.** *Let $n \geq k \geq 1$ and $\mathbf{X} \triangleq \{x_1, \ldots, x_n\} \subset \mathbb{R}^d$. We have that*

$$\mathrm{LOT}_{k,\|\cdot\|_2^2}(\mu_{\mathbf{1}_n, \mathbf{X}}, \mu_{a,\mathbf{X}}) = 2 \min_{Q, z_1, \ldots, z_k} \sum_{i=1}^n \sum_{q=1}^k Q_{i,q} \|x_i - z_q\|_2^2 \ \ s.t. \ Q \in \{0,1\}^{n \times k}, \ Q\mathbf{1}_k = \mathbf{1}_n \ .$$

In the general setting, solving $\mathrm{LOT}_{k,c}(\mu_{a,\mathbf{X}}, \mu_{a,\mathbf{X}})$ for a given geometry $c$, and a prescribed histrogram $a$ offers a new clustering method where the assignment of the points to the clusters is determined by the matrix $Q^*$ solution of (6).

## 6 Computing LOT: Adaptive Stepsizes and Better Initializations

We target in this section practical issues that arises when using [Scetbon et al., 2021, Algo.3] to solve (4). Scetbon et al. [2021] propose to apply a mirror descent scheme with respect to the Kullback-Leibler divergence which boils down to solve at each iteration $k \geq 0$ the following convex problem using the Dykstra's Algorithm [Dykstra, 1983]:

$$(Q_{k+1}, R_{k+1}, g_{k+1}) \triangleq \underset{\zeta \in \mathcal{C}_1(a,b,r) \cap \mathcal{C}_2(r)}{\mathrm{argmin}} \mathrm{KL}(\zeta, \xi_k) \ . \quad (7)$$

where $(Q_0, R_0, g_0) \in \mathcal{C}_1(a,b,r) \cap \mathcal{C}_2(r)$, $\xi_k \triangleq (\xi_k^{(1)}, \xi_k^{(2)}, \xi_k^{(3)})$, $\xi_k^{(1)} \triangleq Q_k \odot \exp(-\gamma_k C R_k \, \mathrm{diag}(1/g_k))$, $\xi_k^{(2)} \triangleq R_k \odot \exp(-\gamma_k C^T Q_k \, \mathrm{diag}(1/g_k))$, $\xi_k^{(3)} \triangleq g_k \odot \exp(\gamma_k \omega_k / g_k^2)$ with $[\omega_k]_i \triangleq [Q_k^T C R_k]_{i,i}$ for all $i \in \{1, \ldots, r\}$, $\mathrm{KL}(\mathbf{w}, \mathbf{r}) \triangleq \sum_i w_i \log(w_i/r_i)$ and $(\gamma_k)_{k\geq 0}$ is a sequence of positive step sizes. In the general setting, each iteration of their algorithm requires $\mathcal{O}(nmr)$ operations and when the ground cost matrix $C$ admits a low-rank factorization of the form $C = AB^T$ where $A \in \mathbb{R}^{n \times q}$ and $B \in \mathbb{R}^{m \times q}$ with $q \ll \min(n, m)$, then the total complexity per iteration becomes linear $\mathcal{O}((n+m)rq)$. Note that for the squared Euclidean cost on $\mathbb{R}^d$, we have that $q = d + 2$. In the following we investigate two practical aspects of the algorithm: the choice of the step sizes and the initialization.

**Adaptive choice of $\gamma_k$.** Scetbon et al. [2021] show experimentally that the choice of $(\gamma_k)_{k\geq 0}$ does not impact the solution obtained upon convergence, but rather the speed at which it is attained. Indeed the larger $\gamma_k$ is, the faster the algorithm will converge. As a result, their algorithm simply relies on a fixed $\gamma$ schedule. However, the range of admissible $\gamma$ depends on the problem considered and it may

vary from one problem to another. Indeed, the algorithm might fail to converge as one needs to ensure at each iteration $k$ of the mirror descent scheme that the kernels $\boldsymbol{\xi}_k$ do not admit 0 entries in order to solve (7) using the Dykstra's Algorithm. Such a situation can occur when the terms involved in the exponentials become too large which may depend on the problem considered. Therefore, it may be of particular interest for practitioners to have a generic range of admissible values for $\gamma$ independently of the considered problem, in order to alleviate parameter tuning issues. We propose to consider instead an adaptive choice of $(\gamma_k)_{k \geq 0}$ along iterations. D'Orazio et al. [2021], Bayandina et al. [2018] have proposed adaptive mirror descent schemes where, at each iteration, the step-size is normalized by the squared dual-norm of the gradient. Applying such a strategy in our case amounts to consider at each iteration

$$\gamma_k = \frac{\gamma}{\| \left( CR\operatorname{diag}(1/g), C^T Q \operatorname{diag}(1/g), -\mathcal{D}(Q^T RC)/g^2 \right) \|_\infty^2}, \tag{8}$$

where the initial $\gamma > 0$ is fixed. By doing so, we are able to guarantee a lower-bound of the exponential terms involved in the expression of the kernels $\boldsymbol{\xi}_k$ at each iteration and prevent them from having 0 entries. We recommend to set such as global $\gamma \in [1, 10]$, and observe that this range works whatever the problem considered.

**On the choice of the initialization.** As $\text{LOT}_{r,c}$ (4) is a non-convex optimization problem, the question of choosing an efficient initialization arises in practice. Scetbon et al. [2021] show experimentally that the convergence of the algorithm does not depend on the initalization chosen if no stopping criterion is used. Indeed, their experimental findings support that only well behaved local minimas are attractive. However, in practice one needs to use a stopping criterion in order to terminate the algorithm. We do observe in many instances that using trivial initializers may result in spurious local minima, which trigger the stopping criterion early on and prevent the algorithm to reach a good solution. Based on various experimentations, we propose to consider a novel initialization of the algorithm. Our initialization aims at being close to a well-behaved local minimum by clustering the input measures. When the measures are supported on Euclidean space, we propose to find $r$ centroids $(z_i)_{i=1}^r$ of one of the two input discrete probability measures using $k$-means and to solve the following convex barycenter problem:

$$\min_{Q,R}\langle C_{X,Z}, Q \rangle + \langle C_{Y,Z}, R \rangle - \varepsilon H(Q) - \varepsilon H(R) \ \text{ s.t. } \ Q\mathbf{1}_n = a, \ R\mathbf{1}_n = b, \ Q^T\mathbf{1}_r = R^T\mathbf{1}_r \,, \tag{9}$$

where $C_{X,Z} = (c(x_i, z_j))_{i,j}$, $C_{Y,Z} = (c(y_i, z_j))_{i,j}$, and $H(P) = -\sum_{i,j} P_{i,j}(\log(P_{i,j} - 1)$. In practice we fix $\varepsilon = 1/10$ and we then initialize $\text{LOT}_{r,c}$ using $(Q, R)$ solution of (9) and $g \triangleq Q^T\mathbf{1}_r (= R^T\mathbf{1}_r)$. Note that $(Q, R, g)$ is an admissible initialization and finding the centroids as well as solving (9) requires $\mathcal{O}((n + m)r)$ algebraic operations. Therefore such initialization does not change the total complexity of the algorithm.

In the general (non-Euclidean) case, we propose to initialize the algorithm by applying our generalized $k$-means approach defined in (6) on each input measure where we fix the common marginal to be $g = \mathbf{1}_r/r$. More precisely, by denoting $C_{X,X} = (c(x_i, x_j))_{i,j}$ and $C_{Y,Y} = (c(y_i, y_j))_{i,j}$, we initialize the algorithm by solving:

$$Q \in \operatorname*{argmin}_{Q}\langle C_{X,X}, Q\operatorname{diag}(1/Q^T\mathbf{1}_n)Q^T \rangle \ \text{ s.t. } \ Q \in \mathbb{R}_+^{n \times k} \,, \ Q\mathbf{1}_k = a, \ Q^T\mathbf{1}_n = \mathbf{1}_r/r \,.$$
$$R \in \operatorname*{argmin}_{R}\langle C_{Y,Y}, R\operatorname{diag}(1/R^T\mathbf{1}_m)R^T \rangle \ \text{ s.t. } \ R \in \mathbb{R}_+^{m \times k} \,, \ R\mathbf{1}_k = b, \ R^T\mathbf{1}_n = \mathbf{1}_r/r \,. \tag{10}$$

Note that again the $(Q, R, g)$ obtained is an admissible initialization and the complexity of solving (10) is of the same order as solving (4), thus the total complexity of the algorithm remains the same.

## 7 Experiments

In this section, we illustrate experimentally our theoretical findings and show how our initialization provide practical improvements. For that purpose we consider 3 synthetic problems and one real world dataset to: *(i)* provide illustrations on the statistical rates of $\text{LOT}_{r,c}$, *(ii)* exhibit the gradient flow of the debiased formulation $\text{DLOT}_{r,c}$, *(iii)* use the clustering method induced by $\text{LOT}_{r,c}$, and *(iv)* show the effect of the initialization. All experiments were run on a MacBook Pro 2019 laptop.

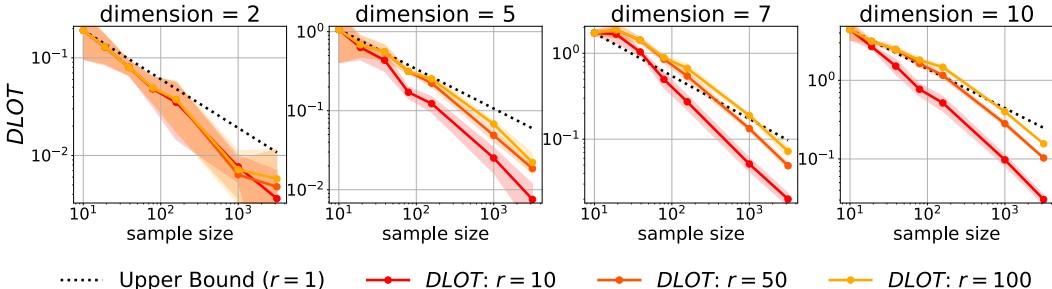

Figure 1: In this experiment, we consider a mixture of 10 anisotropic Gaussians supported on $\mathbb{R}^d$ and we plot the value of $\text{DLOT}_{r,c}$ between two independent empirical measures associated to this mixture when varying the number of samples $n$ and the dimension $d$ for multiple ranks $r$. The ground cost considered is the squared Euclidean distance. Note that $\text{LOT}_r(\mu,\mu) \neq 0$ and therefore we use $\text{DLOT}_{r,c}(\mu,\mu)$ instead to evaluate the rates. Each point has been obtained by repeating 10 times the experiment. We compare the empirical rates obtained with the theoretical one derived in Proposition 4 for $r = 1$. We observe that our theoretical results match the empirical ones and, as expected, the rates do not depend on $d$.

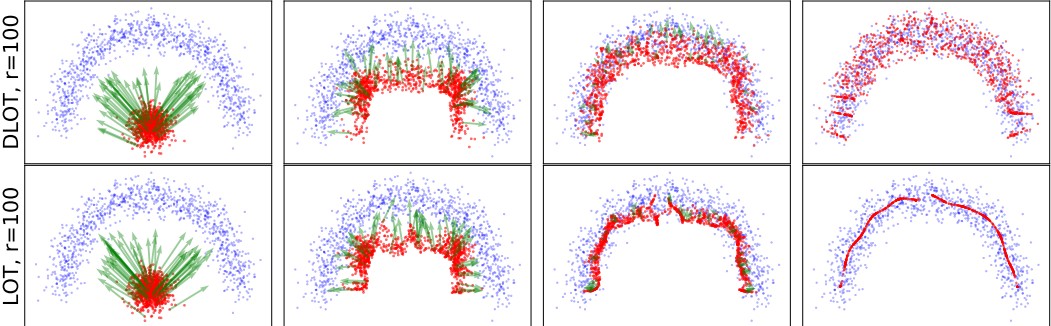

Figure 2: We compare the gradient flows $(\mu_t)_{t\geq 0}$ (in red) starting from a Gaussian distribution, $\mu_0$, to a moon shape distribution (in blue), $\nu$, in 2D when minimizing either $L(\mu) \triangleq \text{DLOT}_{r,c}(\mu,\nu)$ or $L(\mu) \triangleq \text{LOT}_{r,c}(\mu,\nu)$. The ground cost is the squared Euclidean distance and we fix $r = 100$. We consider 1000 samples from each distribution and and we plot the evolution of the probability measure obtained along the iterations of a gradient descent scheme. We also display in green the vector field in the descent direction. We show that the debiased version allows to recover the target distribution while $\text{LOT}_{r,c}$ is learning a biased version with a low-rank structure.

**Statistical rates.** We aim at showing the statistical rates of the plug-in estimator of $\text{LOT}_{r,c}$. As $\text{LOT}_{r,c}(\mu,\mu) \neq 0$ and as we do not have access to this value given samples from $\mu$, we consider instead the debiased version of the low-rank optimal transport, $\text{DLOT}_{r,c}$. In figure 1, we show that the empiricial rates match the theoretical bound obtained in Proposition 4. In particular, we show that that these rates does not depend on the dimension of the ground space. Note also that we recover our theoretical dependence with respect to the rank $r$: the higher the rank, the slower the convergence.

**Gradient Flows using DLOT.** We illustrate here a practical use of DLOT for ML application. In figure 6, we consider $Y_1, \ldots, Y_n$ independent samples from a moon shape distribution in 2D, and by denoting $\hat{\nu}_n$ the empirical measure associated, we show the iterations obtained by a gradient descent scheme on the following optimization problem:

$$\min_{\mathbf{X} \in \mathbb{R}^{n \times 2}} \text{DLOT}_{r,c}(\mu_{\mathbf{1}_n/n, \mathbf{X}}, \hat{\nu}_n) \ .$$

We initialize the algorithm using $n = 1000$ samples drawn from a Gaussian distribution. We show that the gradient flow of our debiased version is able to recover the target distribution. We also compare it with the gradient flow of the biased version (LOT) and show that it fails to reproduce the target distribution as it is learning a biased one with a low-rank structure.

**Application to Clustering.** In this experiment we show some applications of the clustering method induced by $\text{LOT}_{r,c}$. In figure 3, we consider 6 datasets with different structure and we aim at recovering the clusters using (6) for some well chosen costs. We compare the clusters obtained when considering either the squared Euclidean cost (which amounts at applying the $k$-means) and the

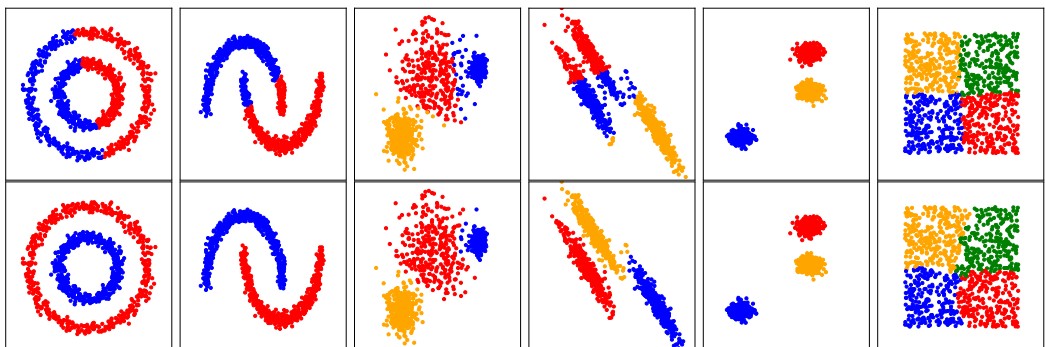

Figure 3: In this experiment, we draw 1000 samples from multiple distributions from the python package scikit-learn [Pedregosa et al., 2011] and we apply the method proposed in (6) for two different costs: in the top row we consider the squared Euclidean distance while in the bottom row, we consider the shortest path distance on the graph associated with the ground cost $c(x, y) = 1 - k(x, y)$ where $k$ is a Gaussian kernel. In the two first problem (starting from the left), we fix $r = 2$, in the next three problem we fix $r = 3$ and in the last one we fix $r = 4$. We observe that the flexibility of our method allows to recover the clustering for a well chosen ground cost.

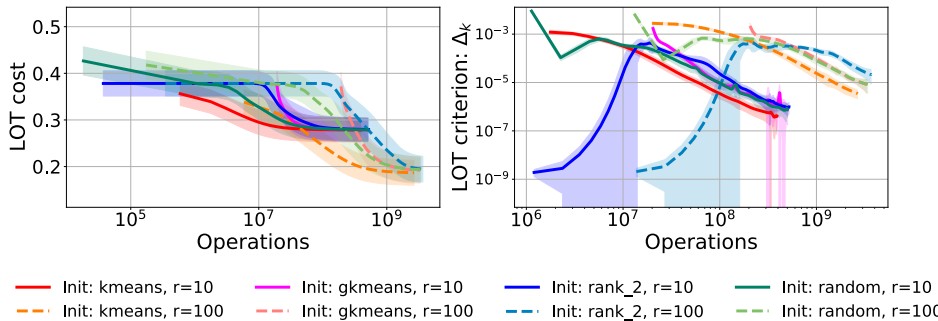

Figure 4: In this experiment, we consider the Newsgroup20 dataset [Pedregosa et al., 2011] constituted of texts and we embed them into distributions in 50D using the same pre-processing steps as in [Cuturi et al., 2022]. We compare different initialization when applying the algorithm of [Scetbon et al., 2021] to compare random texts viewed as distributions for multiple choices of rank $r$. The ground cost considered in the squared Euclidean distance. We repeat the experiments 50 times by sampling randomly multiple problems of similar size ($\simeq 250$ samples). We normalize the cost matrix by its maximum value in order to have comparable LOT cost. We consider 4 different initialization: the one using $k$-means algorithm (9), the one using the generalized $k$-means (10), the rank-2 initialization [Scetbon et al., 2021] and a random initialization where $Q$, $R$ and $g$ are drawn from Gaussians. We compare both the cost value and the criterion value ($\Delta_k$) along the iterations of the MD scheme. Note that the curves obtained do not start at the same point in time as we start plotting the curves after obtaining the initial point which in some case requires more algebraic operations (e.g. kmeans methods). First we observe that whatever the initialization considered, the algorithm converges toward the same value. In addition, we observe that both $k$-means and general $k$-means are able to initialize well the algorithm by avoiding bad local minima at initialization while the two other initialization are close to spurious local minima at initialization.

shortest-path distance on the data viewed as a graph. We show that our method is able to recover the clusters on these settings for well chosen costs and therefore the proposed algorithm in [Scetbon et al. 2021] can be seen as a new alternative in order to clusterize data.

**Effect of the Initialization.** Our goal here is to show the effect of the initialization. In figure 4, we display the evolution of the cost as well as the value of the stopping criterion along the iterations of the MD scheme solving (4) when considering different initialization. The $x$-axis corresponds to the total number of algebraic operations. This number is computed at each iteration of the outer loop of the algorithm proposed in [Scetbon et al.] [2021] and is obtained by computing the complexity of all the operations involved in their algorithm to reach it. We consider this notion of time instead of

CPU/GPU time as we do not want to be architecture/machine dependent. Recall also that the stopping criterion introduced in [Scetbon et al., 2021] is defined for all $k \geq 1$ by

$$\Delta_k \triangleq \frac{1}{\gamma_k^2}(\mathrm{KL}((Q_k, R_k, g_k), (Q_{k-1}, R_{k-1}, g_{k-1})) + \mathrm{KL}((Q_{k-1}, R_{k-1}, g_{k-1}), (Q_k, R_k, g_k))),$$

where $((Q_k, R_k, g_k))_{k \geq 0}$ is the sequence solution of (7). First, we show that whatever the initialization chosen, the algorithm manages to converge to an efficient solution if no stopping criterion is used. However, the choice of the initialization may impact the termination of the algorithm as some initialization might be too close to some spurious local minima. Indeed, the initial points obtained using a "rank 2" or random initialization can be close to spurious and non-attractive local minima, which may trigger the stopping criterion too early and prevent the algorithm from continuing to run in order to converge towards an attractive and well behaved local minimum. We show also that the initialization we propose in (9) and (10) are sufficiently far away from bad local minima and allow the algorithm to converge directly toward the desired solution.

The right figure of Fig.4 shows two main observations: (i) that the initial point obtained using a "rank 2" or random initialization can be close to spurious and non-attractive local minima, which may trigger the stopping criterion too early and prevent the algorithm from continuing to run in order to converge towards an attractive and well behaved local minimum. (ii) When initialiazing the algorithm using kmeans methods, we show that our stopping criterion is a decreasing function of time meaning that the algorithm converges directly towards the desired solution.

**Conclusion.** We assembled in this work theoretical and practical arguments to support low-rank factorizations for OT. We have presented two controls: one concerning the approximation error to the true optimal transport and another concerning the statistical rates of the plug-in estimator. The latter is showed to be independent of the dimension, which is of particular interest when studying OT in ML settings. We have motivated further the use of LOT as a loss by introducing its debiased version and showed that it possesses desirable properties: positivity and metrization of the convergence in law. We have also presented the links between the bias induced by such regularization and clustering methods, and studied empirically the effects of hyperparameters involved in the practical estimation of LOT. The strong theoretical foundations provided in this paper motivate further studies of the empirical behaviour of LOT estimator, notably on finding suitable local minima and on improvements on the convergence of the MD scheme using other adaptive choices for step sizes.

**Acknowledgements.** This work was supported by a "Chaire d'excellence de l'IDEX Paris Saclay". The authors would also like to thank Gabriel Peyré and Jaouad Mourtada for enlightening conversations on the topics discussed in this work.

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
