## Supplementary materials

## A   On the Definition of $\text{LOT}_{r,c}$

Let $(\mathcal{X}, d_{\mathcal{X}})$ and $(\mathcal{Y}, d_{\mathcal{Y}})$ two nonempty compact Polish spaces, $\mu \in \mathcal{M}_1^+(\mathcal{X})$, $\nu \in \mathcal{M}_1^+(\mathcal{Y})$ two probability measures on these spaces and $c : \mathcal{X} \times \mathcal{Y} \to \mathbb{R}_+$ a nonnegative and continuous function. We define the generalized low-rank optimal transport between $\mu$ and $\nu$ as

$$\text{LOT}_{r,c}(\mu, \nu) \triangleq \inf_{\pi \in \Pi_r(\mu,\nu)} \int_{\mathcal{X} \times \mathcal{Y}} c(x, y) d\pi(x, y) \ .$$

where

$$\Pi_r(\mu, \nu) \triangleq \{\pi \in \Pi(\mu, \nu) : \exists (\mu_i)_{i=1}^r \in \mathcal{M}_1^+(\mathcal{X})^r, \ (\nu_i)_{i=1}^r \in \mathcal{M}_1^+(\mathcal{Y})^r, \ \lambda \in \Delta_r^* \ \text{s.t.} \ \pi = \sum_{i=1}^r \lambda_i \mu_i \otimes \nu_i\} \ .$$

As $\mathcal{X}$ and $\mathcal{Y}$ are compact, $\Pi_r(\mu, \nu)$ is tight, then Prokhorov's theorem applies and the closure of $\Pi_r(\mu, \nu)$ is sequentially compact. Let us now show that $\Pi_r(\mu, \nu)$ is closed. Indeed, Let $(\pi_n)_{n \geq 0}$ a sequence of $\Pi_r(\mu, \nu)$ converging towards $\pi_*$. Then by definition there exists for all $k \in [[1, r]]$, $(\mu_n^{(k)})_{n \geq 0}$, $(\nu_n^{(k)})_{n \geq 0}$ and $(\lambda_n^{(k)})_{n \geq 0}$ such that for all $n \geq 0$

$$\pi_n = \sum_{i=1}^r \lambda_n^{(k)} \mu_n^{(k)} \otimes \nu_n^{(k)} \ .$$

However, $(\mu_n^{(k)})_{n \geq 0}$ and $(\nu_n^{(k)})_{n \geq 0}$ are also tight, and Prokhorov's theorem applies, therefore we can extract a common subsequence such that for all $k$,

$$\mu_n^{(k)} \to \mu_*^{(k)} \ \text{and} \ \nu_n^{(k)} \to \nu_*^{(k)}$$

In addition as $(\lambda_n)_{n \geq 0}$ live in the simplex $\Delta_r$, we can also extract a sub-sequence, such that $\lambda_n \to \lambda_* \in \Delta_r$. Finally by unicity of the limit we obtain that

$$\pi_* = \sum_{k=1}^r \lambda_*^{(k)} \mu_*^{(k)} \otimes \nu_*^{(k)} \ .$$

Finally, by denoting $I \triangleq \{k : \lambda_*^{(k)} > 0\}$, and by considering $i^* \in I$, we obtain that

$$\pi_* = \sum_{i \in I \setminus \{i^*\}} \lambda_*^{(i)} \mu_*^{(i)} \otimes \nu_*^{(i)} + \sum_{j=1}^{r-|I|+1} \frac{\lambda_*^{(i^*)}}{r - |I| + 1} \mu_*^{(i^*)} \otimes \nu_*^{(i^*)} \ .$$

from which follows that $\pi_* \in \Pi_r(\mu, \nu)$.

## B   Proofs

### B.1   Proof of Proposition 1

**Proposition.** *Let $n, m \geq 2$, $\mathbf{X} \triangleq \{x_1, \ldots, x_n\} \subset \mathcal{X}$, $\mathbf{Y} \triangleq \{y_1, \ldots, y_m\} \subset \mathcal{Y}$ and $a \in \Delta_n^*$ and $b \in \Delta_m^*$. Then for $2 \leq r \leq \min(n, m)$, we have that*

$$|\text{LOT}_{r,c}(\mu_{a,\mathbf{X}}, \nu_{b,\mathbf{Y}}) - \text{OT}_c(\mu_{a,\mathbf{X}}, \nu_{b,\mathbf{Y}})| \leq \|C\|_\infty \ ln(\min(n, m)/(r - 1))$$

*Proof.* Let $P \in \arg\min_{P \in \Pi_{a,b}} \langle C, P \rangle$. As $P$ is a nonnegative matrix, its nonnegative rank cannot exceed $\min(n, m)$. Assume for simplicity, that $n = m$, then there exists $(R_i)_{i=1}^n$ nonnegative matrices of rank 1 such that

$$P = \sum_{i=1}^n R_i \ .$$

As for all $i \in [|1, n|]$, $R_i$ is a rank 1 matrix, there exist $\tilde{q}_i, \tilde{r}_i \in \mathbb{R}_+^n$ such that $R_i = \tilde{q}_i \tilde{r}_i^T$. Then by denoting $q_i = \tilde{q}_i/|\tilde{q}_i|$, $r_i = \tilde{r}_i/|\tilde{r}_i|$ and $\lambda_i = |\tilde{q}_i||\tilde{r}_i|$ where for any $h \in \mathbb{R}^n$ $|h| \triangleq \sum_{i=1}^n h_i$, we obtain that

$$P = \sum_{i=1}^n \lambda_i q_i r_i^T \ .$$

Without loss of generality, we can consider the case where $\lambda_1 \geq \cdots \geq \lambda_n$. Let us now denote $\lambda := (\lambda_1, \ldots, \lambda_n)$, and by using the fact the $P$ is a coupling we obtain that $\lambda \in \Delta_n$. Also, by definition of $\lambda$, we have that for all $k \in [|1, n|]$, $\lambda_k \leq 1/k$. Let us now define

$$\tilde{P} \triangleq \sum_{i=1}^{r-1} \lambda_i q_i r_i^T + \left(\sum_{i=r}^n \lambda_i\right) \alpha_r \beta_r^T$$

where

$$\alpha_r \triangleq \frac{\sum_{i=r}^n \lambda_i q_i}{\sum_{i=r}^n \lambda_i}$$
$$\beta_r \triangleq \frac{\sum_{i=r}^n \lambda_i r_i}{\sum_{i=r}^n \lambda_i}$$

Remark that $\tilde{P} \in \Pi_{a,b}(r)$, therefore we obtain that

$$|\text{LOT}_{r,c}(\mu_{a,\mathbf{X}}, \nu_{b,\mathbf{Y}}) - \text{OT}_c(\mu_{a,\mathbf{X}}, \nu_{b,\mathbf{Y}})| = \text{LOT}_{r,c}(\mu_{a,\mathbf{X}}, \nu_{b,\mathbf{Y}}) - \text{OT}_x(\mu_{a,\mathbf{X}}, \nu_{b,\mathbf{Y}})$$
$$\leq \langle C, \tilde{P} \rangle - \langle C, P \rangle$$
$$\leq \langle C, \left(\sum_{i=r}^n \lambda_i\right) \alpha_r \beta_r^T \rangle - \langle C, \sum_{i=r}^n \lambda_i q_i r_i^T \rangle$$
$$\leq \langle C, \left(\sum_{i=r}^n \lambda_i\right) \alpha_r \beta_r^T \rangle$$
$$\leq \|C\|_\infty \sum_{i=r}^n \lambda_i \leq \|C\|_\infty \sum_{i=r}^n \frac{1}{i} \leq \|C\|_\infty \ln(n/(r-1))$$

$\square$

## B.2  Proof of Proposition 2

**Proposition 10.** *Let $\mu \in \mathcal{M}_1^+(\mathcal{X})$, $\nu \in \mathcal{M}_1^+(\mathcal{Y})$ and let us assume that $c$ is $L$-Lipschitz w.r.t. $x$ and $y$. Then for any $r \geq 1$, we have*

$$|LOT_{r,c}(\mu, \nu) - OT_c(\mu, \nu)| \leq 2L \max(\mathcal{N}_{\lfloor \log_2(\lfloor \sqrt{r} \rfloor) \rfloor}(\mathcal{X}, d_{\mathcal{X}}), \mathcal{N}_{\lfloor \log_2(\lfloor \sqrt{r} \rfloor) \rfloor}(\mathcal{Y}, d_{\mathcal{Y}}))$$

*Proof.* As $\mathcal{X}$ and $\mathcal{Y}$ are compact, $\mathcal{N}_{\lfloor \log_2(\lfloor \sqrt{r} \rfloor) \rfloor}(\mathcal{X}, d), \mathcal{N}_{\lfloor \log_2(\lfloor \sqrt{r} \rfloor) \rfloor}(\mathcal{Y}, d) < +\infty$ and then by denoting $\varepsilon_{\mathcal{X}} \triangleq \mathcal{N}_{\lfloor \log_2(\lfloor \sqrt{r} \rfloor) \rfloor}(\mathcal{X}, d_{\mathcal{X}})$, there exists $x_1, \ldots, x_{\lfloor \sqrt{r} \rfloor} \in \mathcal{X}$, such that $\mathcal{X} \subset \bigcup_{i=1}^r \mathcal{B}_{\mathcal{X}}(x_i, \varepsilon)$ from which we can extract a partition $(S_{i,\mathcal{X}})_{i=1}^{\lfloor \sqrt{r} \rfloor}$ of $\mathcal{X}$ such that for all $i \in [|1, \lfloor \sqrt{r} \rfloor|]$, and $x, y \in S_{i,\mathcal{X}}$, $d_{\mathcal{X}}(x, y) \leq \varepsilon_{\mathcal{X}}$. Similarly we can build a partition $(S_{i,\mathcal{Y}})_{i=1}^{\lfloor \sqrt{r} \rfloor}$ of $\mathcal{Y}$. Let us now define for all $k \in [|1, \lfloor \sqrt{r} \rfloor|]$,

$$\mu_k \triangleq \frac{\mu|_{S_{k,\mathcal{X}}}}{\mu(S_{k,\mathcal{X}})} \quad \text{and} \quad \nu_k \triangleq \frac{\nu|_{S_{k,\mathcal{Y}}}}{\nu(S_{k,\mathcal{Y}})}$$

with the convention that $\frac{0}{0} = 0$, we can define

$$\pi_r \triangleq \sum_{i,j=1}^{\lfloor \sqrt{r} \rfloor} \pi^*(S_{i,\mathcal{X}} \times S_{j,\mathcal{Y}}) \nu_j \otimes \mu_i \ .$$

First remarks that $\pi_r \in \Pi_r(\mu, \nu)$. Indeed we have for any measurable set $B$

$$\pi_r(\mathcal{X} \times B) = \sum_{j=1}^{\lfloor \sqrt{r} \rfloor^2} \nu_j(B) \sum_{i=1}^{r} \pi^*(S_{i,\mathcal{X}} \times S_{j,\mathcal{Y}})$$

$$= \sum_{j=1}^{\lfloor \sqrt{r} \rfloor} \nu_j(B) \nu(S_{j,\mathcal{Y}})$$

$$= \sum_{j=1}^{\lfloor \sqrt{r} \rfloor} \nu|_{S_{j,\mathcal{X}}}(B)$$

$$= \nu(B) \, ,$$

similarly $\pi_r(A \times \mathcal{Y}) = \mu(A)$ and we have that $\lfloor \sqrt{r} \rfloor^2 \leq r$. Therefore we obtain that

$$|\mathrm{LOT}_{r,c}(\mu, \nu) - \mathrm{OT}_c(\mu, \nu)| = \mathrm{LOT}_{r,c}(\mu, \nu) - \mathrm{OT}_c(\mu, \nu)$$

$$\leq \int_{\mathcal{X} \times \mathcal{Y}} c(x, y) d\pi_r(x, y) - \int_{\mathcal{X} \times \mathcal{Y}} c(x, y) d\pi^*(x, y)$$

$$\leq \sum_{i,j=1}^{\lfloor \sqrt{r} \rfloor} \int_{S_{i,\mathcal{X}} \times S_{j,\mathcal{Y}}} c(x, y) d[\pi_r(x, y) - \pi^*(x, y)]$$

$$\leq \sum_{i,j=1}^{\lfloor \sqrt{r} \rfloor} \pi^*(S_{i,\mathcal{X}} \times S_{j,\mathcal{Y}})$$

$$\times [\sup_{(x,y) \in S_{i,\mathcal{X}} \times S_{j,\mathcal{Y}}} c(x, y) - \inf_{(x,y) \in S_{i,\mathcal{X}} \times S_{j,\mathcal{Y}}} c(x, y)]$$

$$\leq L[\varepsilon_{\mathcal{X}} + \varepsilon_{\mathcal{Y}}]$$

from which the result follows. □

**Corollary.** *Under the same assumptions of Proposition 2 and by assuming in addition that there exists a Monge map solving $OT_c(\mu, \nu)$, we obtain that for any $r \geq 1$,*

$$|\mathrm{LOT}_{r,c}(\mu, \nu) - \mathrm{OT}_c(\mu, \nu)| \leq L\mathcal{N}_{\lfloor \log_2(r) \rfloor}(\mathcal{Y}, d_{\mathcal{Y}})$$

*Proof.* Let us denote $T$ a Monge map solution of $\mathrm{OT}_c(\mu, \nu)$ and as in the proof above, let us consider a partition of $(S_{i,\mathcal{Y}})_{i=1}^{r}$ of $\mathcal{Y}$ such that for all $i \in [[1, r]]$, and $x, y \in S_{i,\mathcal{Y}}$, $d_{\mathcal{Y}}(x, y) \leq \varepsilon_{\mathcal{Y}}$ with $\varepsilon_{\mathcal{Y}} \triangleq \mathcal{N}_{\lfloor \log_2(r) \rfloor}(\mathcal{Y}, d_{\mathcal{Y}})$. Let us now define for all $k \in [[1, \lfloor \sqrt{r} \rfloor]]$,

$$\mu_k \triangleq \frac{\mu|_{T^{-1}(S_{k,\mathcal{Y}})}}{\mu(T^{-1}(S_{k,\mathcal{Y}}))} \text{ and } \nu_k \triangleq \frac{\nu|_{S_{k,\mathcal{Y}}}}{\nu(S_{k,\mathcal{Y}})}$$

with the convention that $\frac{0}{0} = 0$, we can define

$$\pi_r \triangleq \sum_{k=1}^{r} \pi^*(T^{-1}(S_{k,\mathcal{Y}}) \times S_{k,\mathcal{Y}}) \nu_k \otimes \mu_k \, .$$

Again we have that $\pi_r \in \Pi_r(\mu, \nu)$, and we obtain that

$$|\text{LOT}_{r,c}(\mu, \nu) - \text{OT}_c(\mu, \nu)| = \text{LOT}_{r,c}(\mu, \nu) - \text{OT}_c(\mu, \nu)$$

$$\leq \int_{\mathcal{X} \times \mathcal{Y}} c(x,y) d\pi_r(x,y) - \int_{\mathcal{X} \times \mathcal{Y}} c(x,y) d\pi^*(x,y)$$

$$\leq \sum_{k=1}^{r} \pi^*(T^{-1}(S_{k,\mathcal{Y}}) \times S_{k,\mathcal{Y}}) \int_{T^{-1}(S_{k,\mathcal{Y}}) \times S_{k,\mathcal{Y}}} c(x,y) d\mu_k(y) \otimes \nu_k(y)$$

$$- \sum_{k=1}^{r} \int_{T^{-1}(S_{k,\mathcal{Y}})} c(x, T(x)) d\mu(x)$$

$$\leq \sum_{k=1}^{r} \pi^*(T^{-1}(S_{k,\mathcal{Y}}) \times S_{k,\mathcal{Y}}) \int_{T^{-1}(S_{k,\mathcal{Y}}) \times S_{k,\mathcal{Y}}} c(x,y) d\mu_k(y) \otimes \nu_k(y)$$

$$- \sum_{k=1}^{r} \pi^*(T^{-1}(S_{k,\mathcal{Y}}) \times S_{k,\mathcal{Y}}) \int_{T^{-1}(S_{k,\mathcal{Y}}) \times S_{k,\mathcal{Y}}} c(x, T(x)) d\mu_k(x) \otimes \nu_k(y)$$

$$\leq \sum_{k=1}^{r} \pi^*(T^{-1}(S_{k,\mathcal{Y}}) \times S_{k,\mathcal{Y}}) \int_{T^{-1}(S_{k,\mathcal{Y}}) \times S_{k,\mathcal{Y}}} [c(x,y) - c(x, T(x))] d\mu_k(x) \otimes \nu_k(x)$$

$$\leq L\varepsilon_{\mathcal{Y}}$$

from which the result follows. Note that to obtain the above inequalities, we use the fact that $\pi^*$ is supported on the graph of $T$, and therefore we have have for all $k \in [|1, r|]$,

$$\pi^*(T^{-1}(S_{k,\mathcal{Y}}) \times S_{k,\mathcal{Y}}) = \mu(T^{-1}(S_{k,\mathcal{Y}})) = \nu(S_{k,\mathcal{Y}}).$$

$\square$

### B.3 Proof of Proposition 3

**Proposition.** *Let $r \geq 1$ and $\mu, \nu \in \mathcal{M}_1^+(\mathcal{X})$, then $\text{LOT}_{r,c}(\hat{\mu}_n, \hat{\nu}_n) \xrightarrow[n \to +\infty]{} \text{LOT}_{r,c}(\mu, \nu)$ a.s.*

*Proof.* Let $\pi^*$ solution of $\text{LOT}_{r,c}(\mu, \nu)$. Then there exists $\lambda^* \in \Delta_r^*$, $(\mu_i^*)_{i=1}^r$, $(\nu_i^*)_{i=1}^r \in \mathcal{M}_1^+(\mathcal{X})^r$ such that

$$\pi^* = \sum_{i=1}^{r} \lambda_i^* \mu_i^* \otimes \nu_i^*.$$

Note that by definition, we have that

$$\mu = \sum_{i=1}^{r} \lambda_i^* \mu_i^* \quad \text{and} \quad \nu = \sum_{i=1}^{r} \lambda_i^* \nu_i^*.$$

Let us now define $\pi_\mu$ and $\pi_\mu$ both elements of $\mathcal{M}_1^+(\mathcal{X} \times [|1, r|])$ as follows:

$\pi_\mu(A \times \{k\}) \triangleq \lambda_k \mu_k(A)$ and $\pi_\nu(A \times \{k\}) \triangleq \lambda_k \nu_k(A)$ for any measurable set $A$ and $k \in [|1, r|]$.
Observe that the right marginals of $\pi_\mu$ and $\pi_\nu$ is the same and we will denote it $\rho$. We can now define for all $x, y \in \mathcal{X}$ the family of kernels $(k_\mu(\cdot, x))_{x \in \mathcal{X}} \in \mathcal{M}_1^+([|1, r|])^{\mathcal{X}}$ and $(k_\nu(\cdot, y))_{y \in \mathcal{X}} \in \mathcal{M}_1^+([|1, r|])^{\mathcal{X}}$ corresponding to the disintegration with respect to the projection of respectively $\mu$ and $\nu$. Let us now consider $n$ independent samples $(Z_i^\mu)_{i=1}^n$ and $(Z_i^\nu)_{i=1}^n$ such that for all $i \in [|1, n|]$, $Z_i^\mu \sim k_\mu(\cdot, X_i)$ and $Z_i^\nu \sim k_\nu(\cdot, Y_i)$ and let us define for all $k \in [|1, r|]$

$$\tilde{\mu}_k \triangleq \frac{1}{n} \sum_{i=1}^{n} \mathbf{1}_{Z_i^\mu = k} \delta_{X_i} \quad \text{and} \quad \tilde{\nu}_k \triangleq \frac{1}{n} \sum_{i=1}^{n} \mathbf{1}_{Z_i^\nu = k} \delta_{Y_i}.$$

Let us now define

$$\tilde{\pi} \triangleq \sum_{k=1}^{r-1} \frac{\min(|\tilde{\mu}_k|, |\tilde{\nu}_k|)}{|\tilde{\mu}_k||\tilde{\nu}_k|} \tilde{\mu}_k \otimes \tilde{\nu}_k$$

$$+ \frac{1}{1 - \sum_{k=1}^{r-1} \min(|\tilde{\mu}_k|, |\tilde{\nu}_k|)} \left[\hat{\mu} - \sum_{k=1}^{r-1} \frac{\min(|\tilde{\mu}_k|, |\tilde{\nu}_k|)}{|\tilde{\mu}_k|} \tilde{\mu}_k\right] \otimes \left[\hat{\nu} - \sum_{k=1}^{r-1} \frac{\min(|\tilde{\mu}_k|, |\tilde{\nu}_k|)}{|\tilde{\nu}_k|} \tilde{\nu}_k\right]$$

with the convention that $\frac{0}{0} = 0$. Now it is easy to check that $\tilde{\pi} \in \Pi_r(\hat{\mu}, \hat{\nu})$, indeed we have that

$$\tilde{\pi}(A \times \mathcal{X}) = \sum_{k=1}^{r-1} \frac{\min(|\tilde{\mu}_k|, |\tilde{\nu}_k|)}{|\tilde{\mu}_k|} \tilde{\mu}_k(A)$$

$$+ \frac{1}{1 - \sum_{k=1}^{r-1} \min(|\tilde{\mu}_k|, |\tilde{\nu}_k|)} \left[ \hat{\mu}(A) - \sum_{k=1}^{r-1} \frac{\min(|\tilde{\mu}_k|, |\tilde{\nu}_k|)}{|\tilde{\mu}_k|} \tilde{\mu}_k(A) \right] \left[ 1 - \sum_{k=1}^{r-1} \min(|\tilde{\mu}_k|, |\tilde{\nu}_k|) \right]$$

$$= \hat{\mu}(A)$$

in addition by construction we have that

$$\left| \hat{\mu} - \sum_{k=1}^{r-1} \frac{\min(|\tilde{\mu}_k|, |\tilde{\nu}_k|)}{|\tilde{\mu}_k|} \tilde{\mu}_k \right| = \left| \hat{\nu} - \sum_{k=1}^{r-1} \frac{\min(|\tilde{\mu}_k|, |\tilde{\nu}_k|)}{|\tilde{\nu}_k|} \tilde{\nu}_k \right| = 1 - \sum_{k=1}^{r-1} \min(|\tilde{\mu}_k|, |\tilde{\nu}_k|)$$

and both $\hat{\mu} - \sum_{k=1}^{r-1} \frac{\min(|\tilde{\mu}_k|, |\tilde{\nu}_k|)}{|\tilde{\mu}_k|} \tilde{\mu}_k$ and $\hat{\nu} - \sum_{k=1}^{r-1} \frac{\min(|\tilde{\mu}_k|, |\tilde{\nu}_k|)}{|\tilde{\nu}_k|} \tilde{\nu}_k$ are positive measures. Therefore we obtain that

$$\text{LOT}_{r,c}(\hat{\mu}, \hat{\nu}) \leq \int_{\mathcal{X}^2} c(x, y) d\tilde{\pi}(x, y)$$

Now we aim at showing at $\int_{\mathcal{X}^2} c(x, y) d\tilde{\pi}(x, y) \to \text{LOT}_{r,c}(\mu, \nu)$ $a.s.$. Indeed first observe that from the law of large numbers we have that for all $k \in [[1, r]]$, $|\tilde{\mu}_k| \to \lambda_k^*$ and similarly $|\tilde{\nu}_k| \to \lambda_k^*$. In addition, for all $k, q$ we have that almost surely, $\tilde{\mu}_k \otimes \tilde{\nu}_q$ converges weakly towards $\lambda_k^* \lambda_q^* \mu_k \otimes \nu_q$. Indeed one can consider the following algebra $\mathcal{F} \triangleq \left\{ (x, y) \in \mathcal{X}^2 \to f(x)g(y) \ f, g \in \mathcal{C}(\mathcal{X}) \right\}$, and then by Stone-Weierstrass, one obtains by density the desired result. Now remark that

$$\int_{\mathcal{X}^2} c(x, y) d\tilde{\pi}(x, y) = \sum_{k=1}^{r-1} \frac{\min(|\tilde{\mu}_k|, |\tilde{\nu}_k|)}{|\tilde{\mu}_k||\tilde{\nu}_k|} \int_{\mathcal{X}^2} c(x, y) d\tilde{\mu}_k \otimes \tilde{\nu}_k$$

$$+ \frac{1}{\tilde{\lambda}_r} \int_{\mathcal{Z}^2} c(x, y) d\tilde{\mu}_r \otimes \tilde{\nu}_r$$

$$+ \frac{1}{\tilde{\lambda}_r} \sum_{k=1}^{r-1} \left( 1 - \frac{\min(|\tilde{\mu}_k|, |\tilde{\nu}_k|)}{|\tilde{\nu}_k|} \right) \int_{\mathcal{X}^2} c(x, y) d\tilde{\mu}_r \otimes \tilde{\nu}_k$$

$$+ \frac{1}{\tilde{\lambda}_r} \sum_{k=1}^{r-1} \left( 1 - \frac{\min(|\tilde{\mu}_k|, |\tilde{\nu}_k|)}{|\tilde{\mu}_k|} \right) \int_{\mathcal{X}^2} c(x, y) d\tilde{\mu}_k \otimes \tilde{\nu}_r$$

$$+ \frac{1}{\tilde{\lambda}_r} \sum_{k,q=1}^{r-1} \int_{\mathcal{X}^2} \left( 1 - \frac{\min(|\tilde{\mu}_k|, |\tilde{\nu}_k|)}{|\tilde{\mu}_k|} \right) \left( 1 - \frac{\min(|\tilde{\mu}_q|, |\tilde{\nu}_q|)}{|\tilde{\nu}_q|} \right) c(x, y) d\tilde{\mu}_k(x) d\tilde{\nu}_q(y)$$

from which follows directly that $\int_{\mathcal{X}^2} c(x, y) d\tilde{\pi}(x, y) \to \text{LOT}_{r,c}(\mu, \nu)$ $a.s.$. Let us now denote for all $n \geq 1$, $\pi_n$ a solution of $\text{LOT}_{r,c}(\hat{\mu}, \hat{\nu})$. Let $\omega \in \Omega$ an element of the probability space where live the random variables $(X_i)_{i \geq 0}$ and $(Y_i)_{i \geq 0}$ such that $\int_{\mathcal{X}^2} c(x, y) d\tilde{\pi}^{(\omega)}(x, y) \to \text{LOT}_{r,c}(\mu, \nu)$. As $\mathcal{X}$ is compact Thanks to Prokhorov's Theorem, we can extract a sequence such that $(\pi_n^{(\omega)})_{n \geq 0}$ converge weakly towards $\pi^{(\omega)} \in \Pi_r(\mu, \nu)$. In addition we have that for all $n \geq 1$

$$\int_{\mathcal{X}^2} c(x, y) d\pi_n^{(\omega)}(x, y) \leq \int_{\mathcal{X}^2} c(x, y) d\tilde{\pi}^{(\omega)}(x, y)$$

And by considering the limit we obtain that

$$\int c(x, y) d\pi^{(\omega)}(x, y) \leq \text{LOT}_{r,c}(\mu, \nu)$$

However $\pi^{(\omega)} \in \Pi_r(\mu, \nu)$ and by optimality we obtain that

$$\int c(x, y) d\pi^{(\omega)}(x, y) = \text{LOT}_{r,c}(\mu, \nu)$$

This holds for an arbitrary subsequence of $(\pi_n^{(\omega)})_{n \geq 0}$, from which follows that $\int c(x, y) d\pi_n^{(\omega)}(x, y) \to \text{LOT}_{r,c}(\mu, \nu)$. Finally this holds almost surely and the result follows. $\qquad \square$

## B.4 Proof of Proposition 4

**Proposition.** *Let $r \geq 1$ and $\mu, \nu \in \mathcal{M}_1^+(\mathcal{X})$. Then, there exists a constant $K_r$ such that for any $\delta > 0$ and $n \geq 1$, we have, with a probability of at least $1 - 2\delta$, that*

$$\mathrm{LOT}_{r,c}(\hat{\mu}_n, \hat{\nu}_n) - \mathrm{LOT}_{r,c}(\mu, \nu) \leq 11\|c\|_\infty \sqrt{\frac{r}{n}} + K_r \|c\|_\infty \left[ \sqrt{\frac{\log(40/\delta)}{n}} + \frac{\sqrt{r}\log(40/\delta)}{n} \right]$$

*Proof.* We reintroduce the same notation as in the proof of Proposition 3. Let $\pi^*$ solution of $\mathrm{LOT}_{r,c}(\mu, \nu)$. Then there exists $\lambda^* \in \Delta_r^*$, $(\mu_i^*)_{i=1}^r$, $(\nu_i^*)_{i=1}^r \in \mathcal{M}_1^+(\mathcal{Z})^r$ such that

$$\pi^* = \sum_{i=1}^r \lambda_i^* \mu_i^* \otimes \nu_i^*.$$

As before let us also consider $\pi_\mu$ and $\pi_\mu$ defined as $\pi_\mu(A \times \{k\}) \triangleq \lambda_k \mu_k(A)$ and $\pi_\nu(A \times \{k\}) \triangleq \lambda_k \nu_k(A)$ for any measurable set $A$ and $k \in [\![1, r]\!]$ and denote $\rho$ their common right marginal. We also consider $n$ independent samples $(Z_i^\mu)_{i=1}^n$ and $(Z_i^\nu)_{i=1}^n$ such that for all $i \in [\![1, n]\!]$, $Z_i^\mu \sim k_\mu(\cdot, X_i)$ and $Z_i^\nu \sim k_\nu(\cdot, Y_i)$ and we denote for all $k \in [\![1, r]\!]$

$$\tilde{\mu}_k \triangleq \frac{1}{n} \sum_{i=1}^n \mathbf{1}_{Z_i^\mu = k} \delta_{X_i} \quad \text{and} \quad \tilde{\nu}_k \triangleq \frac{1}{n} \sum_{i=1}^n \mathbf{1}_{Z_i^\nu = k} \delta_{Y_i}$$

Let us now define

$$\hat{\pi} \triangleq \sum_{i=1}^r \frac{1}{\lambda_k^*} \tilde{\mu}_k \otimes \tilde{\nu}_k .$$

Our goal is to control the following quantity:

$$\left| \mathrm{LOT}_{r,c}(\mu, \nu) - \int_{\mathcal{Z}^2} c(x, y) d\hat{\pi}(x, y) \right|,$$

First observe that

$$\mathbb{E}\left[ \int_{\mathcal{Z}^2} c(x, y) d\hat{\pi}(x, y) \right] = \sum_{i=1}^r \frac{1}{\lambda_k^*} \mathbb{E}\left[ \int_{\mathcal{Z}^2} c(x, y) d\tilde{\mu}_k(x) d\tilde{\nu}_k(y) \right]$$

$$= \sum_{i=1}^r \frac{1}{\lambda_k^* n^2} \times \sum_{i,j} \mathbb{E}\left[ c(X_i, Y_j) \mathbf{1}_{Z_i^\mu = k} \mathbf{1}_{Z_j^\nu = k} \right]$$

Moreover, we have that

$$\mathbb{E}\left[ c(X_i, Y_j) \mathbf{1}_{Z_i^\mu = k} \mathbf{1}_{Z_j^\nu = k} \right] = \int_{(\mathcal{Z} \times [\![1, r]\!])^2} c(x, y) \mathbf{1}_{z=k} \mathbf{1}_{z'=k} d\pi_\mu(x, z) d\pi_\nu(y, z')$$

$$= \int_{(\mathcal{Z} \times [\![1, r]\!])^2} c(x, y) \mathbf{1}_{z=k} \mathbf{1}_{z'=k} d\mu_z(x) d\nu_{z'}(y) d\rho(z) d\rho(z')$$

$$= \lambda_k^2 \int_{\mathcal{Z}^2} c(x, y) d\mu_k(x) d\nu_k(y)$$

from which follows that

$$\mathbb{E}\left[ \int_{\mathcal{Z}^2} c(x, y) d\hat{\pi}(x, y) \right] = \sum_{i=1}^r \lambda_k^* \int_{\mathcal{Z}^2} c(x, y) d\mu_k(x) d\nu_k(y) = \mathrm{LOT}_{r,c}(\mu, \nu)$$

Now let us define for all $(x_i, z_i)_{i=1}^n, (y_i, z_i') \in (\mathcal{Z} \times [\![1, r]\!])^n$,

$$g((x_1, z_1), \ldots, (x_n, z_n), (y_1, z_1'), \ldots, (y_n, z_n')) \triangleq \sum_{q=1}^r \frac{1}{\lambda_q^* n^2} \sum_{i,j} c(x_i, y_j) \mathbf{1}_{z_i = q} \mathbf{1}_{z_j' = q} ,$$

since $\mathcal{Z}$ is compact and $c$ is continuous, we have that

$$|g(\dots, (x_k, z_k), \dots) - g(\dots, (\tilde{x}_k, \tilde{z}_k), \dots)| = \left| \sum_{q=1}^{r} \frac{1}{\lambda_q^* n^2} \sum_j [c(x_k, y_j) \mathbf{1}_{z_k=q} - c(\tilde{x}_k, y_j) \mathbf{1}_{\tilde{z}_k=q}] \mathbf{1}_{z_j'=q} \right|$$

$$= \left| \frac{1}{\lambda_{z_k}^* n^2} \sum_{j=1}^{n} c(x_k, y_j) \mathbf{1}_{z_j'=z_k} - \frac{1}{\lambda_{\tilde{z}_k}^* n^2} \sum_{j=1}^{n} c(\tilde{x}_k, y_j) \mathbf{1}_{z_j'=\tilde{z}_k} \right|$$

$$\leq \frac{\|c\|_\infty}{n^2} \left[ \frac{\sum_{j=1}^{n} \mathbf{1}_{z_j'=z_k}}{\lambda_{z_k}^*} + \frac{\sum_{j=1}^{n} \mathbf{1}_{z_j'=\tilde{z}_k}}{\lambda_{z_k}^*} \right]$$

$$\leq \frac{2\|c\|_\infty}{\min\limits_{1\leq q\leq r} \lambda_q^*} \frac{1}{n}$$

Then by applying the McDiarmid's inequality we obtain that for $\delta > 0$, with a probability at least of $1 - \delta$, we have

$$\left| \text{LOT}_{r,c}(\mu, \nu) - \int_{\mathcal{Z}^2} c(x, y) d\hat{\pi}(x, y) \right| \leq \frac{2\|c\|_\infty}{\min\limits_{1\leq q\leq r} \lambda_q^*} \sqrt{\frac{\log(2/\delta)}{n}}$$

Now we aim at building a coupling $\tilde{\pi} \in \Pi_r(\hat{\mu}, \hat{\nu})$ from $\hat{\pi}$. Let us consider the same as the one introduce in the proof of Proposition B.3, that is

$$\tilde{\pi} \triangleq \sum_{k=1}^{r-1} \frac{\min(|\tilde{\mu}_k|, |\tilde{\nu}_k|)}{|\tilde{\mu}_k||\tilde{\nu}_k|} \tilde{\mu}_k \otimes \tilde{\nu}_k$$

$$+ \frac{1}{1 - \sum_{k=1}^{r-1} \min(|\tilde{\mu}_k|, |\tilde{\nu}_k|)} \left[ \hat{\mu} - \sum_{k=1}^{r-1} \frac{\min(|\tilde{\mu}_k|, |\tilde{\nu}_k|)}{|\tilde{\mu}_k|} \tilde{\mu}_k \right] \otimes \left[ \hat{\nu} - \sum_{k=1}^{r-1} \frac{\min(|\tilde{\mu}_k|, |\tilde{\nu}_k|)}{|\tilde{\nu}_k|} \tilde{\nu}_k \right]$$

with the convention that $\frac{0}{0} = 0$. Let us now expand the above expression, and by denoting $\tilde{\lambda}_r = 1 - \sum_{k=1}^{r-1} \min(|\tilde{\mu}_k|, |\tilde{\nu}_k|)$ we obtain that

$$\tilde{\pi} = \sum_{k=1}^{r-1} \frac{\min(|\tilde{\mu}_k|, |\tilde{\nu}_k|)}{|\tilde{\mu}_k||\tilde{\nu}_k|} \tilde{\mu}_k \otimes \tilde{\nu}_k$$

$$+ \frac{1}{\tilde{\lambda}_r} \tilde{\mu}_r \otimes \tilde{\nu}_r$$

$$+ \frac{1}{\tilde{\lambda}_r} \tilde{\mu}_r \otimes \left[ \sum_{k=1}^{r-1} \left( 1 - \frac{\min(|\tilde{\mu}_k|, |\tilde{\nu}_k|)}{|\tilde{\nu}_k|} \right) \tilde{\nu}_k \right]$$

$$+ \frac{1}{\tilde{\lambda}_r} \left[ \sum_{k=1}^{r-1} \left( 1 - \frac{\min(|\tilde{\mu}_k|, |\tilde{\nu}_k|)}{|\tilde{\mu}_k|} \right) \tilde{\mu}_k \right] \otimes \tilde{\nu}_r$$

$$+ \frac{1}{\tilde{\lambda}_r} \left[ \sum_{k=1}^{r-1} \left( 1 - \frac{\min(|\tilde{\mu}_k|, |\tilde{\nu}_k|)}{|\tilde{\mu}_k|} \right) \tilde{\mu}_k \right] \otimes \left[ \sum_{k=1}^{r-1} \left( 1 - \frac{\min(|\tilde{\mu}_k|, |\tilde{\nu}_k|)}{|\tilde{\nu}_k|} \right) \tilde{\nu}_k \right]$$

Now we aim at controlling the following quantity $\left|\int_{\mathcal{Z}^2} c(x,y) d\hat{\pi}(x,y) - \int_{\mathcal{Z}^2} c(x,y) d\tilde{\pi}(x,y)\right|$ and we observe that

$$\int_{\mathcal{Z}^2} c(x,y) d[\hat{\pi}(x,y) - \tilde{\pi}(x,y)] = \sum_{k=1}^{r-1} \int_{\mathcal{Z}^2} c(x,y) \left[ \frac{1}{\lambda_k^*} - \frac{\min(|\tilde{\mu}_k|,|\tilde{\nu}_k|)}{|\tilde{\mu}_k||\tilde{\nu}_k|} \right] d\tilde{\mu}_k(x)\tilde{\nu}_k(y) \quad (11)$$

$$+ \int_{\mathcal{Z}^2} c(x,y) \left[ \frac{1}{\lambda_r^*} - \frac{1}{\tilde{\lambda}_r} \right] d\tilde{\mu}_r(x)\tilde{\nu}_r(y) \quad (12)$$

$$+ \frac{1}{\tilde{\lambda}_r} \sum_{k=1}^{r-1} \int_{\mathcal{Z}^2} \left( 1 - \frac{\min(|\tilde{\mu}_k|,|\tilde{\nu}_k|)}{|\tilde{\nu}_k|} \right) c(x,y) d\tilde{\mu}_r(x) d\tilde{\nu}_k(y) \quad (13)$$

$$+ \frac{1}{\tilde{\lambda}_r} \sum_{k=1}^{r-1} \int_{\mathcal{Z}^2} \left( 1 - \frac{\min(|\tilde{\mu}_k|,|\tilde{\nu}_k|)}{|\tilde{\mu}_k|} \right) c(x,y) d\tilde{\mu}_k(x) d\tilde{\nu}_r(y) \quad (14)$$

$$+ \frac{1}{\tilde{\lambda}_r} \sum_{k,q=1}^{r-1} \int_{\mathcal{Z}^2} \left( 1 - \frac{\min(|\tilde{\mu}_k|,|\tilde{\nu}_k|)}{|\tilde{\mu}_k|} \right) \left( 1 - \frac{\min(|\tilde{\mu}_q|,|\tilde{\nu}_q|)}{|\tilde{\nu}_q|} \right) c(x,y) d\tilde{\mu}_k(x) d\tilde{\nu}_q(y) \quad (15)$$

Let us now control each term of the RHS of the above equality. Let us first consider the term in Eq. 11, remark that we have

$$\left| \int_{\mathcal{Z}^2} c(x,y) \left[ \frac{1}{\lambda_k^*} - \frac{\min(|\tilde{\mu}_k|,|\tilde{\nu}_k|)}{|\tilde{\mu}_k||\tilde{\nu}_k|} \right] d\tilde{\mu}_k(x)\tilde{\nu}_k(y) \right|$$

$$\leq \left| \left[ \frac{1}{\lambda_k^*} - \frac{\min(|\tilde{\mu}_k|,|\tilde{\nu}_k|)}{|\tilde{\mu}_k||\tilde{\nu}_k|} \right] \right| \|c\|_\infty |\tilde{\mu}_k||\tilde{\nu}_k|$$

$$\leq \left| \left[ \frac{|\tilde{\mu}_k||\tilde{\nu}_k|}{\lambda_k^*} - \min(|\tilde{\mu}_k|,|\tilde{\nu}_k|) \right] \right| \|c\|_\infty$$

$$\leq \min(|\tilde{\mu}_k|,|\tilde{\nu}_k|) \left| \frac{\max(|\tilde{\mu}_k|,|\tilde{\nu}_k|)}{\lambda_k^*} - 1 \right| \|c\|_\infty$$

$$\leq \frac{\min(|\tilde{\mu}_k|,|\tilde{\nu}_k|)}{\lambda_k^*} |\max(|\tilde{\mu}_k|,|\tilde{\nu}_k|) - \lambda_k^*| \|c\|_\infty$$

$$\leq \frac{\min(|\tilde{\mu}_k|,|\tilde{\nu}_k|)}{\lambda_k^*} \max(\|\tilde{\lambda}_\mu - \lambda^*\|_\infty, \|\tilde{\lambda}_\nu - \lambda^*\|_\infty)\|c\|_\infty$$

$$\leq \|c\|_\infty \max \left( \left\| \frac{\tilde{\lambda}_\mu}{\lambda^*} \right\|_\infty, \left\| \frac{\tilde{\lambda}_\nu}{\lambda^*} \right\|_\infty \right) \max(\|\tilde{\lambda}_\mu - \lambda^*\|_\infty, \|\tilde{\lambda}_\nu - \lambda^*\|_\infty)$$

where we have denoted $\tilde{\lambda}_\mu \triangleq (|\tilde{\mu}_k|)_{k=1}^r$ and $\tilde{\lambda}_\nu \triangleq (|\tilde{\nu}_k|)_{k=1}^r$. Now observe that

$$\mathbb{P}\left( \max(\|\tilde{\lambda}_\mu - \lambda^*\|_\infty, \|\tilde{\lambda}_\nu - \lambda^*\|_\infty) \geq t \right) \leq 2\mathbb{P}\left( \|\tilde{\lambda}_\mu - \lambda^*\|_\infty \geq t \right)$$

$$\leq \mathbb{P}\left( d_K(\lambda^*, \tilde{\lambda}_\mu) \geq \frac{t}{2} \right)$$

$$\leq 4\exp(-nt^2/2)$$

where $d_K$ is the Kolmogorov distance. In addition we have

$$\max \left( \left\| \frac{\tilde{\lambda}_\mu}{\lambda^*} \right\|_\infty, \left\| \frac{\tilde{\lambda}_\nu}{\lambda^*} \right\|_\infty \right) \leq 1 + \frac{1}{\min_{1\leq i\leq r} \lambda_i^*} \max \left( \|\tilde{\lambda}_\mu - \lambda^*\|_\infty, \|\tilde{\lambda}_\nu - \lambda^*\|_\infty \right)$$

Combining the two above controls, we obtain that for all $\delta > 0$, with a probability of at least $1 - \delta$,

$$\left| \int_{\mathcal{Z}^2} c(x,y) \left[ \frac{1}{\lambda_k^*} - \frac{\min(|\tilde{\mu}_k|,|\tilde{\nu}_k|)}{|\tilde{\mu}_k||\tilde{\nu}_k|} \right] d\tilde{\mu}_k(x)\tilde{\nu}_k(y) \right| \leq \|c\|_\infty \sqrt{\frac{2\ln 8/\delta}{n}} + \frac{\|c\|_\infty}{n} \frac{2\ln 8/\delta}{\min_{1\leq i\leq r} \lambda_i^*}$$

Let us now consider the term in Eq. 12, we have that

$$\left| \int_{\mathcal{Z}^2} c(x,y) \left[ \frac{1}{\lambda_r^*} - \frac{1}{\tilde{\lambda}_r} \right] d\tilde{\mu}_r(x) \tilde{\nu}_r(y) \right| \leq \frac{|\tilde{\mu}_r||\tilde{\nu}_r|}{\lambda_r^* \tilde{\lambda}_r} \left| 1 - \sum_{i=1}^{r} \min(|\tilde{\mu}_k|, |\tilde{\nu}_k|) - \lambda_r \right| \|c\|_\infty$$

$$\leq \max \left( \left\| \frac{\tilde{\lambda}_\mu}{\lambda^*} \right\|_\infty, \left\| \frac{\tilde{\lambda}_\nu}{\lambda^*} \right\|_\infty \right) \sum_{k=1}^{r-1} |\lambda_k^* - \min(|\tilde{\mu}_k|, |\tilde{\nu}_k|)| \, \|c\|_\infty$$

$$\leq \max \left( \left\| \frac{\tilde{\lambda}_\mu}{\lambda^*} \right\|_\infty, \left\| \frac{\tilde{\lambda}_\nu}{\lambda^*} \right\|_\infty \right) \|c\|_\infty (\|\lambda^* - \tilde{\lambda}_\mu\|_1 + \|\lambda^* - \tilde{\lambda}_\nu\|_1)$$

$$\leq 2\|c\|_\infty \max \left( \left\| \frac{\tilde{\lambda}_\mu}{\lambda^*} \right\|_\infty, \left\| \frac{\tilde{\lambda}_\nu}{\lambda^*} \right\|_\infty \right) \max(\|\lambda^* - \tilde{\lambda}_\mu\|_1, \|\lambda^* - \tilde{\lambda}_\nu\|_1)$$

However we have that

$$\mathbb{P}\left( \max(\|\lambda^* - \tilde{\lambda}_\mu\|_1, \|\lambda^* - \tilde{\lambda}_\nu\|_1) \geq t \right) \leq 2\mathbb{P}\left( \|\lambda^* - \tilde{\lambda}_\mu\|_1 \geq t \right)$$

In addition we have that $\mathbb{E}(\|\lambda^* - \tilde{\lambda}_\mu\|_1) \leq \sqrt{\frac{r}{n}}$ and by applying the McDiarmid's Inequality, we obtain that for all $\delta > 0$, with a probability of $1 - \delta$

$$\|\lambda^* - \tilde{\lambda}_\mu\|_1 \leq \sqrt{\frac{r}{n}} + \sqrt{\frac{2\ln(2/\delta)}{n}}$$

Therefore we obtain that with a probability of at least $1 - \delta$,

$$\left| \int_{\mathcal{Z}^2} c(x,y) \left[ \frac{1}{\lambda_r^*} - \frac{1}{\tilde{\lambda}_r} \right] d\tilde{\mu}_r(x) \tilde{\nu}_r(y) \right| \leq 2\|c\|_\infty \left[ \sqrt{\frac{r}{n}} + \sqrt{\frac{2\ln(8/\delta)}{n}} + \frac{2\ln(8/\delta) + \sqrt{2r\ln(8/\delta)}}{n \times \min_{1 \leq i \leq r} \lambda_i^*} \right]$$

For the term in Eq. 13 and 14, we obtain that

$$\left| \frac{1}{\tilde{\lambda}_r} \sum_{k=1}^{r-1} \int_{\mathcal{Z}^2} \left( 1 - \frac{\min(|\tilde{\mu}_k|, |\tilde{\nu}_k|)}{|\tilde{\nu}_k|} \right) c(x,y) d\tilde{\mu}_r(x) d\tilde{\nu}_k(y) \right|$$

$$\leq \frac{|\tilde{\mu}_r|}{\tilde{\lambda}_r} \sum_{k=1}^{r-1} (|\tilde{\nu}_k| - \min(|\tilde{\mu}_k|, |\tilde{\nu}_k|)) \|c\|_\infty$$

$$\leq \frac{|\tilde{\mu}_r|}{\tilde{\lambda}_r} [\tilde{\lambda}_r - |\tilde{\nu}_r|] \|c\|_\infty$$

$$\leq [|\tilde{\lambda}_r - \lambda_r^*| + |\lambda_r^* - \tilde{\nu}_r|] \|c\|_\infty$$

$$\leq 3\|c\|_\infty \max(\|\lambda^* - \tilde{\lambda}_\mu\|_1, \|\lambda^* - \tilde{\lambda}_\nu\|_1)$$

Therefore we obtain that with a probability of at least $1 - \delta$,

$$\left| \frac{1}{\tilde{\lambda}_r} \sum_{k=1}^{r-1} \int_{\mathcal{Z}^2} \left( 1 - \frac{\min(|\tilde{\mu}_k|, |\tilde{\nu}_k|)}{|\tilde{\nu}_k|} \right) c(x,y) d\tilde{\mu}_r(x) d\tilde{\nu}_k(y) \right| \leq 3\|c\|_\infty \left[ \sqrt{\frac{r}{n}} + \sqrt{\frac{2\ln(2/\delta)}{n}} \right]$$

Finally the last term in Eq. 15 can be controlled as the following:

$$\left| \frac{1}{\tilde{\lambda}_r} \sum_{k,q=1}^{r-1} \int_{\mathcal{Z}^2} \left( 1 - \frac{\min(|\tilde{\mu}_k|, |\tilde{\nu}_k|)}{|\tilde{\mu}_k|} \right) \left( 1 - \frac{\min(|\tilde{\mu}_q|, |\tilde{\nu}_q|)}{|\tilde{\nu}_q|} \right) c(x,y) d\tilde{\mu}_k(x) d\tilde{\nu}_q(y) \right|$$

$$\leq \frac{\|c\|_\infty}{\tilde{\lambda}_r} \sum_{k,q=1}^{r-1} \left( 1 - \frac{\min(|\tilde{\mu}_k|, |\tilde{\nu}_k|)}{|\tilde{\mu}_k|} \right) \left( 1 - \frac{\min(|\tilde{\mu}_q|, |\tilde{\nu}_q|)}{|\tilde{\nu}_q|} \right) |\tilde{\mu}_k||\tilde{\nu}_q|$$

$$\leq \frac{\|c\|_\infty}{\tilde{\lambda}_r} \sum_{k=1}^{r-1} (|\tilde{\mu}_k| - \min(|\tilde{\mu}_k|, |\tilde{\nu}_k|)) \sum_{k=1}^{r-1} (|\tilde{\nu}_k| - \min(|\tilde{\mu}_k|, |\tilde{\nu}_k|))$$

$$\leq 3\|c\|_\infty \max(\|\lambda^* - \tilde{\lambda}_\mu\|_1, \|\lambda^* - \tilde{\lambda}_\nu\|_1)$$

and we obtain that with a probability of at least $1 - \delta$,

$$\left| \frac{1}{\tilde{\lambda}_r} \sum_{k,q=1}^{r-1} \int_{\mathcal{Z}^2} \left( 1 - \frac{\min(|\tilde{\mu}_k|, |\tilde{\nu}_k|)}{|\tilde{\mu}_k|} \right) \left( 1 - \frac{\min(|\tilde{\mu}_q|, |\tilde{\nu}_q|)}{|\tilde{\nu}_q|} \right) c(x,y) d\tilde{\mu}_k(x) d\tilde{\nu}_q(y) \right|$$

$$\leq 3\|c\|_\infty \left[ \sqrt{\frac{r}{n}} + \sqrt{\frac{2\ln(2/\delta)}{n}} \right]$$

Then by applying a union bound we obtain that with a probability of at least $1 - \delta$

$$\left| \int_{\mathcal{Z}^2} c(x,y) d[\hat{\pi}(x,y) - \tilde{\pi}(x,y)] \right| \leq \|c\|_\infty \left[ 11\sqrt{\frac{r}{n}} + 12\sqrt{\frac{2\ln 40/\delta}{n}} + \frac{6\ln(40/\delta) + 2\sqrt{2r\ln(40/\delta)}}{n \times \min\limits_{1 \leq i \leq r} \lambda_i^*} \right]$$

Now observe that

$$\text{LOT}_{r,c}(\hat{\mu}, \hat{\nu}) - \text{LOT}_{r,c}(\mu, \nu) \leq \int_{\mathcal{Z}^2} c(x,y) d\tilde{\pi}(x,y) - \int_{\mathcal{Z}^2} c(x,y) d\pi^*(x,y)$$

$$\leq \int_{\mathcal{Z}^2} c(x,y) d[\tilde{\pi} - \hat{\pi}](x,y) + \int_{\mathcal{Z}^2} c(x,y) d[\hat{\pi} - \pi^*](x,y)$$

and by combining the two control we obtain that with a probability of at least $1 - 2\delta$,

$$\text{LOT}_{r,c}(\hat{\mu}, \hat{\nu}) - \text{LOT}_{r,c}(\mu, \nu) \leq \|c\|_\infty \left[ 11\sqrt{\frac{r}{n}} + 12\sqrt{\frac{2\ln 40/\delta}{n}} + \frac{1}{\alpha} \left( 2\sqrt{\frac{\log(2/\delta)}{n}} + \frac{6\ln(40/\delta) + 2\sqrt{2r\ln(40/\delta)}}{n} \right) \right]$$

$$\leq 11\|c\|_\infty \sqrt{\frac{r}{n}} + \frac{14\|c\|_\infty}{\alpha} \sqrt{\frac{\log(40/\delta)}{n}} + \frac{2\|c\|_\infty \max(6, \sqrt{2r}) \log(40/\delta)}{n\alpha}$$

where $\alpha \triangleq \min\limits_{1 \leq i \leq r} \lambda_i^*$ and the result follows. $\qquad\square$

## B.5 Proof Proposition 5

**Proposition.** *Let $r \geq 1$, $\delta > 0$ and $\mu, \nu \in \mathcal{M}_1^+(\mathcal{X})$. Then there exists a constant $N_{r,\delta}$ such that if $n \geq N_{r,\delta}$ then with a probability of at least $1 - 2\delta$, we have*

$$\text{LOT}_{r,c}(\hat{\mu}_n, \hat{\nu}_n) - \text{LOT}_{r,c}(\mu, \nu) \leq 11\|c\|_\infty \sqrt{\frac{r}{n}} + 77\|c\|_\infty \sqrt{\frac{\log(40/\delta)}{n}} .$$

*Proof.* We consider the same notations as in the proof of Proposition 4. In particular let us define for all $(x_i, z_i)_{i=1}^n, (y_i, z_i') \in (\mathcal{Z} \times [\![1,r]\!])^n$,

$$g((x_1, z_1), \ldots, (x_n, z_n), (y_1, z_1'), \ldots, (y_n, z_n')) \triangleq \sum_{q=1}^r \frac{1}{\lambda_q^* n^2} \sum_{i,j} c(x_i, y_j) \mathbf{1}_{z_i=q} \mathbf{1}_{z_j'=q} ,$$

Recall that we have

$$|g(\ldots, (x_k, z_k), \ldots) - g(\ldots, (\tilde{x}_k, \tilde{z}_k), \ldots)| \leq \frac{\|c\|_\infty}{n^2} \left[ \frac{\sum_{j=1}^n \mathbf{1}_{z_j'=z_k}}{\lambda_{z_k}^*} + \frac{\sum_{j=1}^n \mathbf{1}_{z_j'=\tilde{z}_k}}{\lambda_{\tilde{z}_k}^*} \right]$$

$$\leq \frac{2\|c\|_\infty}{n} \max \left( \left\| \frac{\tilde{\lambda}_\mu}{\lambda^*} \right\|_\infty, \left\| \frac{\tilde{\lambda}_\nu}{\lambda^*} \right\|_\infty \right)$$

$$\leq \frac{2\|c\|_\infty}{n} + \frac{2\|c\|_\infty}{n \times \min\limits_{1 \leq i \leq r} \lambda_i^*} \max \left( \|\tilde{\lambda}_\mu - \lambda^*\|_\infty, \|\tilde{\lambda}_\nu - \lambda^*\|_\infty \right)$$

In fact if we have a control in probability of the bounded difference we can use an extension of the McDiarmid's Inequality. For that purpose let us first introduce the following definition.

**Definition 4.** *Let $(X_i)_{i=1}^m$, $m$ independent random variables and $g$ a measurable function. We say that $g$ is weakly difference-bounded with respect to $(X_i)_{i=1}^m$ by $(b, \beta, \delta)$ if*

$$\mathbb{P}\left(|g(X_1, \ldots, X_m) - g(X_1', \ldots, X_m')| \leq \beta\right) \geq 1 - \delta$$

*with $X_i' = X_i$ except for one coordinate $k$ where $X_k'$ is an independent copy of $X_k$. Furthermore for any $(x_i)_{i=1}^m$ and $(x_i')_{i=1}^m$ where for all coordinate except on $x_j = x_j'$*

$$|g(x_1, \ldots, x_m) - g(x_1', \ldots, x_m')| \leq b .$$

Let us now introduce an extension of McDiarmid's Inequality [Kutin, 2002].

**Theorem 1.** *Let $(X_i)_{i=1}^m$, $m$ independent random variables and $g$ a measurable function which is weakly difference-bounded with respect to $(X_i)_{i=1}^m$ by $(b, \beta/m, \exp(-Km))$, then if $0 < \tau \leq T(b, \beta, K)$ and $m \geq M(b, \beta, K, \tau)$, then*

$$\mathbb{P}(|g(X_1, \ldots, X_m) - \mathbb{E}(g(X_1, \ldots, X_m))| \geq \tau) \leq 4 \exp\left(\frac{-\tau^2 m}{8\beta^2}\right)$$

*where*

$$T(b, \beta, K) \triangleq \min\left(\frac{14c}{2}, 4\beta\sqrt{K}, \frac{\beta^2 K}{b}\right)$$

$$M(b, \beta, K, \tau) \triangleq \max\left(\frac{b}{\beta}, \beta\sqrt{40}, 3\left(\frac{24}{K} + 3\right)\log\left(\frac{24}{K} + 3\right), \frac{1}{\tau}\right)$$

Given the above Theroem we can obtain an asymptotic control of the deviation of $g$ from its mean. Let $\delta' > 0$ and let us denote

$$m \triangleq 2n$$

$$b \triangleq \frac{2\|c\|_\infty}{n \times \min\limits_{1 \leq i \leq r} \lambda_i^*}$$

$$K \triangleq \frac{\log(1/\delta')}{2n}$$

$$\beta \triangleq 4\|c\|_\infty \left[1 + \frac{1}{\min\limits_{1 \leq i \leq r} \lambda_i^*}\sqrt{\frac{2\log(4/\delta')}{n}}\right]$$

Observe now that with a probability of at least $1 - \exp(-Km)$

$$|g(\ldots, (x_k, z_k), \ldots) - g(\ldots, (\tilde{x}_k, \tilde{z}_k), \ldots)| \leq \frac{2\|c\|_\infty}{n}\left[1 + \frac{1}{\min\limits_{1 \leq i \leq r} \lambda_i^*}\sqrt{\frac{2\log(4/\delta')}{n}}\right]$$

Let us now fix $\delta > 0$ and let us choose $\delta'$ such that $\delta' \triangleq 4/n$ and $\tau \triangleq \beta\sqrt{\frac{4\log(4/\delta)}{n}}$, then we obtain that for $n$ sufficiently large (such that $n \geq M(b, \beta, K, \tau)/2$ and $\tau \leq T(b, \beta, K)$), we have that with a probability of at least $1 - \delta$

$$\left|\text{LOT}_{r,c}(\mu, \nu) - \int_{\mathcal{Z}^2} c(x, y) d\hat{\pi}(x, y)\right| \leq 4\|c\|_\infty \left[1 + \frac{1}{\min\limits_{1 \leq i \leq r} \lambda_i^*}\sqrt{\frac{2\log(n)}{n}}\right]\sqrt{\frac{4\log(4/\delta)}{n}}$$

$$\leq 4\|c\|_\infty \sqrt{\frac{4\log(4/\delta)}{n}} + \frac{16\sqrt{5}\|c\|_\infty \sqrt{\log(n)\log(4/\delta)}}{n \times \min\limits_{1 \leq i \leq r} \lambda_i^*}$$

Recall also from the proof of Proposition 4, that we have with a probability of at least $1 - \delta$

$$\left|\int_{\mathcal{Z}^2} c(x, y) d[\hat{\pi}(x, y) - \tilde{\pi}(x, y)]\right| \leq \|c\|_\infty \left[11\sqrt{\frac{r}{n}} + 12\sqrt{\frac{2\ln 40/\delta}{n}} + \frac{6\ln(40/\delta) + 2\sqrt{2r\ln(40/\delta)}}{n \times \min\limits_{1 \leq i \leq r} \lambda_i^*}\right]$$

Finally by imposing in addition that

$$\sqrt{\frac{n}{\log(n)}} \geq \frac{1}{\min\limits_{1\leq i \leq r} \lambda_i^*} \quad, \quad \sqrt{n} \geq \frac{\sqrt{\log(40/\delta)}}{\min\limits_{1\leq i \leq r} \lambda_i^*} \quad \text{and} \quad \sqrt{n} \geq \frac{\sqrt{r}}{\min\limits_{1\leq i \leq r}}$$

we obtain that for $n$ is large enough (such that (such that $n \geq M(b,\beta,K,\tau)/2$ and $\tau \leq T(b,\beta,K)$) and satysfing the above inequalities, we have with a probability of at least $1 - 2\delta$ that

$$\text{LOT}_{r,c}(\hat{\mu},\hat{\nu}) - \text{LOT}_{r,c}(\mu,\nu) \leq 11\|c\|_\infty \sqrt{\frac{r}{n}} + 77\|c\|_\infty \sqrt{\frac{\log(40/\delta)}{n}}$$

$\square$

## B.6 Proof Proposition 6

**Proposition.** *Let $\mu,\nu \in \mathcal{M}_1^+(\mathcal{X})$. Let us assume that $c$ is symmetric, then we have*

$$\text{DLOT}_{1,c}(\mu,\nu) = \frac{1}{2}\int_{\mathcal{X}^2} -c(x,y)d[\mu-\nu] \otimes d[\mu-\nu](x,y) \ .$$

*If in addition we assume the $c$ is Lipschitz w.r.t to $x$ and $y$, then we have*

$$\text{DLOT}_{r,c}(\mu,\nu) \xrightarrow[r\to+\infty]{} \text{OT}_c(\mu,\nu) \ .$$

*Proof.* When $r = 1$, it is clear that for any $\mu,\nu \in \mathcal{M}_1^+(\mathcal{X})$, $\Pi_r(\mu,\nu) = \{\mu \otimes \nu\}$ and thanks to the symmetry of $c$, we have directly that

$$\text{DLOT}_{1,c}(\mu,\nu) = \frac{1}{2}\int_{\mathcal{X}^2} -c(x,y)d[\mu-\nu] \otimes d[\mu-\nu](x,y) = \frac{1}{2}\text{MMD}_{-c}(\mu,\nu) \ .$$

The limit is a direct consequence of Proposition 2. $\square$

## B.7 Proof of Proposition 8

**Proposition.** *Let $r \geq 1$ and $(\mu_n)_{n\geq 0}$ and $(\nu_n)_{n\geq 0}$ two sequences of probability measures such that $\mu_n \to \mu$ and $\nu_n \to \nu$ with respect to the convergence in law. Then we have that*

$$LOT_{r,c}(\mu_n,\nu_n) \to LOT_{r,c}(\mu,\nu) \ .$$

*Proof.* Let us denote $\pi$ an optimal solution of $\text{LOT}_{r,c}(\mu,\nu)$ and let us denote $(\mu^{(i)})_{i=1}^r$, $(\nu^{(i)})_{i=1}^r$ and $(\lambda^{(i)})_{i=1}^r$ the decomposition associated. In the following Lemma, we aim at building specific decompositions of the sequences $(\mu_n)_{n\geq 0}$ and $(\nu_n)_{n\geq 0}$.

**Lemma 1.** *Let $r \geq 1$, $\mu \in \mathcal{M}_1^+(\mathcal{X})$ and $(\mu^{(i)})_{i=1}^r \in \mathcal{M}_1^+(\mathcal{X})$ and $(\lambda^{(i)})_{i=1}^r \in \Delta_r^*$ such that $\mu = \sum_{i=1}^r \lambda_i \mu^{(i)}$. Then for any sequence of probability measures $(\mu_n)_{\geq 0}$ such that $\mu_n \to \mu$, there exist for all $i \in [|1,r|]$ a sequence of nonnegative measures $(\mu_n^{(i)})_{n\geq 0}$ such that*

$$\mu_n^{(i)} \to \lambda_i \mu^{(i)} \text{ for all } i \in [|1,r|] \text{ and}$$

$$\sum_{i=1}^r \mu_n^{(i)} = \mu_n \text{ for all } n \geq 0$$

*Proof.* For $r = 1$ the result is clear. Let us now show the result for $r = 2$. Let us denote $(\tilde{\mu}_n^{(1)})$ a sequence converging weakly towards $\lambda_1 \mu^{(1)}$. Then by denoting $\mu_n^{(1)} \triangleq \mu_n - (\mu_n - \tilde{\mu}_n^{(1)})_+$ where $(\cdot)_+$ correspond to the non-negative part of the measure, we have that

$$\mu_n^{(1)} \geq 0, \ \mu_n^{(1)} \to \lambda_1 \mu^{(1)},$$
$$\mu_n^{(2)} \triangleq \mu_n - \mu_n^{(1)} \geq 0, \ \mu_n^{(2)} \to \lambda_2 \mu^{(2)} \text{ and}$$
$$\mu_n = \mu_n^{(1)} + \mu_n^{(2)} \text{ for all } n \geq 0$$

which is the result. Let $r \geq 2$ and let us assume that the result holds for all $1 \leq k \leq r$. Let us now consider a decomposition of $\mu$ such that $\mu = \sum_{i=1}^{r+1} \lambda_i \mu^{(i)}$. By denoting $\tilde{\mu}^{(1)} \triangleq \frac{\sum_{i=1}^{r} \lambda_i \mu^{(i)}}{\sum_{i=1}^{r} \lambda_i}$, we obtain that

$$\mu = \left( \sum_{i=1}^{r} \lambda_i \right) \tilde{\mu}^{(1)} + \lambda_{r+1} \mu^{(r+1)} .$$

Then by recursion we have that there exists sequences of nonnegative measures $(\tilde{\mu}_n^{(1)})$ and $(\mu_n^{(r+1)})$ such that

$$\tilde{\mu}_n^{(1)} \to \left( \sum_{i=1}^{r} \lambda_i \right) \tilde{\mu}^{(1)}, \ \mu_n^{(r+1)} \to \lambda_{r+1} \mu^{(r+1)} \ \text{and} \ \mu_n = \tilde{\mu}_n^{(1)} + \mu_n^{(r+1)} \ \text{for all} \ n \geq 0$$

Now observe that $\frac{\tilde{\mu}_n^{(1)}}{|\tilde{\mu}_n^{(1)}|} \to \tilde{\mu}^{(1)} = \sum_{i=1}^{r} \frac{\lambda_i}{\sum_{i=1}^{r} \lambda_i} \mu^{(i)}$. Therefore applying the recursion on this problem allows us to obtain a decomposition of $\tilde{\mu}_n^{(1)}$ of the form

$$\frac{\tilde{\mu}_n^{(1)}}{|\tilde{\mu}_n^{(1)}|} = \sum_{i=1}^{r} \mu_n^{(i)} \ \text{where}$$

$$\mu_n^{(i)} \geq 0 \ \text{and} \ \mu_n^{(i)} \to \frac{\lambda_i}{\sum_{i=1}^{r} \lambda_i} \mu^{(i)} .$$

Therefore we obtain that

$$\mu_n = \sum_{i=1}^{r} |\tilde{\mu}_n^{(1)}| \mu_n^{(i)} + \mu_n^{(r+1)} \ \text{where}$$

$$\mu_n^{(i)} \geq 0, \ |\tilde{\mu}_n^{(1)}| \mu_n^{(i)} \to \lambda_i \mu^{(i)} \ \text{for all} \ i \in [|1, r|] \ \text{and}$$

$$\mu_n^{(r+1)} \geq 0, \ \mu_n^{(r+1)} \to \lambda_{r+1} \mu^{(r+1)}$$

from which follows the result. □

Let us now consider such decompositions of $(\mu_n)_{n \geq 0}$ and $(\nu_n)_{n \geq 0}$ such that each factor converges toward the target decomposition of $\mu$. Now let us build the following coupling:

$$\tilde{\pi}_n \triangleq \sum_{k=1}^{r-1} \frac{\min(|\mu_n^{(k)}|, |\nu_n^{(k)}|)}{|\mu_n^{(k)}||\nu_n^{(k)}|} \mu_n^{(k)} \otimes \mu_n^{(k)}$$

$$+ \frac{1}{1 - \sum_{k=1}^{r-1} \min(|\mu_n^{(k)}|, |\nu_n^{(k)}|)} \left[ |\mu_n| - \sum_{k=1}^{r-1} \frac{\min(|\mu_n^{(k)}|, |\nu_n^{(k)}|)}{|\mu_n^{(k)}|} \mu_n^{(k)} \right] \otimes \left[ \nu_n - \sum_{k=1}^{r-1} \frac{\min(|\mu_n^{(k)}|, |\nu_n^{(k)}|)}{|\nu_n^{(k)}|} \nu_n^{(k)} \right]$$

with the convention that $\frac{0}{0} = 0$. Now it is easy to check that $\tilde{\pi}_n \in \Pi_r(\mu_n, \nu_n)$, and we have that

$$\mathrm{LOT}_{r,c}(\mu_n, \nu_n) \leq \int_{\mathcal{X}^2} d(x, y) d\tilde{\pi}_n(x, y) \to \mathrm{LOT}_{r,c}(\mu, \nu)$$

and by Prokhorov's theorem and the optimality of the limit of $(\tilde{\pi}_n)_{n \geq 0}$ (up to an extraction) we obtain that $\mathrm{LOT}_{r,c}(\mu_n, \nu_n) \to \mathrm{LOT}_{r,c}(\mu, \nu)$. □

## B.8 Proof Proposition 7

**Proposition.** *Let $r \geq 1$, and let us assume that $c$ is a semimetric of negative type. Then for all $\mu, \nu \in \mathcal{M}_1^+(\mathcal{X})$, we have that*

$$DLOT_r(\mu, \nu) \geq 0 .$$

*In addition, if $c$ has strong negative type then we have also that*

$$\mathrm{DLOT}_{r,c}(\mu, \nu) = 0 \iff \mu = \nu \ \text{and}$$

$$\mu_n \to \mu \iff \mathrm{DLOT}_{r,c}(\mu_n, \mu) \to 0 .$$

*where the convergence of the sequence of probability measures considered is the convergence in law.*

*Proof.* Let $\pi^*$ solution of $\text{LOT}_{r,c}(\mu, \nu)$. Then there exists $\lambda^* \in \Delta_r^*$, $(\mu_i^*)_{i=1}^r$, $(\nu_i^*)_{i=1}^r \in \mathcal{M}_1^+(\mathcal{X})^r$ such that

$$\pi^* = \sum_{i=1}^r \lambda_i^* \mu_i^* \otimes \nu_i^*.$$

Note that by definition, we have that

$$\mu = \sum_{i=1}^r \lambda_i^* \mu_i^* \text{ and } \nu = \sum_{i=1}^r \lambda_i^* \nu_i^*,$$

By definition we have also that

$$\text{LOT}_{r,c}(\mu, \mu) \leq \sum_{k=1}^r \lambda_k^* \int_{\mathcal{X}^2} c(x,y) d\mu_k^* \otimes \mu_k^*$$

similarly for $\text{LOT}_{r,c}(\nu, \nu)$ we have

$$\text{LOT}_{r,c}(\nu, \nu) \leq \sum_{k=1}^r \lambda_k^* \int_{\mathcal{X}^2} c(x,y) d\nu_k^* \otimes \nu_k^*$$

Therefore we have

$$\text{DLOT}_{r,c}(\mu, \nu) \geq \sum_{k=1}^r \lambda_k^* \left( \int_{\mathcal{X}^2} c(x,y) d\mu_k^* \otimes \nu_k^* - \frac{1}{2} \left[ \int_{\mathcal{X}^2} c(x,y) d\mu_k^* \otimes \mu_k^* + \int_{\mathcal{X}^2} c(x,y) d\nu_k^* \otimes \nu_k^* \right] \right)$$

$$\geq \sum_{k=1}^r \lambda_k^* \int_{\mathcal{X}^2} -c(x,y) d[\mu_k^* - \nu_k^*] \otimes [\mu_k^* - \nu_k^*]$$

$$\geq \sum_{k=1}^r \frac{\lambda_k^*}{2} D_c(\mu_k^*, \nu_k^*)$$

where for any any probability measures $\alpha, \beta$ on $\mathcal{X}$ we define

$$D_c(\alpha, \beta) \triangleq 2 \int_{\mathcal{X}^2} c(x,y) d\alpha \otimes \beta - \int_{\mathcal{X}^2} c(x,y) d\alpha \otimes \alpha - \int_{\mathcal{X}^2} c(x,y) d\beta \otimes \beta$$

However, as $c$ is assumed to have a negative type, we have that

$$D_c(\mu_k^*, \nu_k^*) \geq 0 \ \forall k$$

In addition if we assume that $c$ has a strong negative type, then we obtain directly that

$$\text{DLOT}_{r,c}(\mu, \nu) = 0 \implies \mu_k^* = \nu_k^* \ \forall k .$$

Let us now show that $\text{DLOT}_{r,c}$ metrize the convergence in law. The direct implication is a direct consequence of the Proposition 8. Conversely, if $\text{DLOT}_{r,c}(\mu_n, \mu) \to 0$, then by compacity of $\mathcal{X}$ and thanks to the Prokhorov's theorem we can extract a subsequence of $\mu_n \to \mu^*$, and thanks to Proposition 8, we also obtain that $\text{DLOT}_{r,c}(\mu_n, \mu) \to \text{DLOT}_{r,c}(\mu^*, \mu)$. Finally we deduce that $\text{DLOT}_{r,c}(\mu^*, \mu) = 0$ and $\mu^* = \mu$.

$\square$

## B.9 Proof Proposition 9

**Proposition.** *Let $n \geq k \geq 1$, $\mathbf{X} \triangleq \{x_1, \ldots, x_n\} \subset \mathcal{X}$ and $a \in \Delta_n^*$. If $c$ is a semimetric of negative type, then by denoting $C = (c(x_i, x_j))_{i,j}$, we have that*

$$\text{LOT}_{k,c}(\mu_{a,\mathbf{X}}, \mu_{a,\mathbf{X}}) = \min_Q \langle C, Q diag(1/Q^T \mathbf{1}_n) Q^T \rangle \ s.t. \ Q \in \mathbb{R}_+^{n \times k} , \ Q \mathbf{1}_k = a . \tag{16}$$

*Proof.* First remarks that one can reformulate the $\text{LOT}_{k,c}$ problem as

$$\text{LOT}_{k,c}(\mu, \mu) \triangleq \min_{g \in \Delta_k^*} \min_{(\mathbf{x},\mathbf{y}) \in K_{a,g}^2} \sum_{i=1}^k \frac{\mathbf{x}_i^T C \mathbf{y}_i}{g_i}$$

where

$$K_{a,g} \triangleq \{\mathbf{x} \in \mathbb{R}^{nk} \text{ s.t. } A\mathbf{x} = [a,g]^T, \ \mathbf{x} \geq 0\}$$

$$A \triangleq \begin{pmatrix} \mathbf{1}_n^T \otimes \mathbb{I}_k \\ \mathbb{I}_n^T \otimes \mathbf{1}_k \end{pmatrix} \text{ and }$$

$$\mathbf{x}_i \triangleq [x_{(i-1)\times n+1}, \ldots, x_{i\times n}]^T, \ \mathbf{y}_i \triangleq [y_{(i-1)\times n+1}, \ldots, y_{i\times n}]^T \text{ for all } i \in [|1,k|]$$

Indeed the above optimization problem is just a reformulation of $\text{LOT}_{k,c}(\mu,\mu)$ where we have vectorized the couplings in a column-wise order. Let us now show the following lemma from which the result will follow.

**Lemma 2.** *Under the same assumption of Proposition 9 we have that for all $g \in \Delta_k^*$*

$$\min_{(\mathbf{x},\mathbf{y}) \in K_{a,g}^2} \sum_{i=1}^{k} \frac{\mathbf{x}_i^T C \mathbf{y}_i}{g_i} = \min_{\mathbf{x} \in K_{a,g}} \sum_{i=1}^{k} \frac{\mathbf{x}_i^T C \mathbf{x}_i}{g_i}$$

*Proof.* Let $(\mathbf{x}^*, \mathbf{y}^*)$ solution of the LHS optimization problem. Then we have that

$$\sum_{i=1}^{k} \frac{(\mathbf{x}_i^*)^T C \mathbf{x}_i^*}{g_i} \geq \sum_{i=1}^{k} \frac{(\mathbf{x}_i^*)^T C \mathbf{y}_i^*}{g_i}$$

$$\sum_{i=1}^{k} \frac{(\mathbf{y}_i^*)^T C \mathbf{y}_i^*}{g_i} \geq \sum_{i=1}^{k} \frac{(\mathbf{x}_i^*)^T C \mathbf{y}_i^*}{g_i}$$

Therefore we obtain that

$$0 \leq \sum_{i=1}^{k} \frac{(\mathbf{x}_i^*)^T C \mathbf{x}_i^*}{g_i} - \sum_{i=1}^{k} \frac{(\mathbf{x}_i^*)^T C \mathbf{y}_i^*}{g_i} = \sum_{i=1}^{k} \frac{(\mathbf{x}_i^*)^T C (\mathbf{x}_i^* - \mathbf{y}_i^*)}{g_i}$$

$$0 \leq \sum_{i=1}^{k} \frac{(\mathbf{y}_i^*)^T C \mathbf{y}_i^*}{g_i} - \sum_{i=1}^{k} \frac{(\mathbf{x}_i^*)^T C \mathbf{y}_i^*}{g_i} = \sum_{i=1}^{k} \frac{(\mathbf{y}_i^* - \mathbf{x}_i^*)^T C \mathbf{y}_i^*}{g_i}$$

Then by symmetry of $C$, we obtain by adding the two terms that

$$\sum_{i=1}^{k} \frac{(\mathbf{x}_i^* - \mathbf{y}_i^*)^T C (\mathbf{x}_i^* - \mathbf{y}_i^*)}{g_i} \geq 0$$

However, thanks to the linear constraints, we have that for all $i \in [|1,k|]$,

$$\sum_{q=0}^{n-1} x_{(i-1)\times n+1+q}^* = \sum_{q=0}^{n-1} y_{(i-1)\times n+1+q}^* = g_i$$

Therefore $(\mathbf{x}_i^* - \mathbf{y}_i^*)^T \mathbf{1}_n = 0$ and thanks to the negativity of the cost function $c$ we obtain that

$$(\mathbf{x}_i^* - \mathbf{y}_i^*)^T C (\mathbf{x}_i^* - \mathbf{y}_i^*) \leq 0$$

Therefore we have that

$$(\mathbf{x}_i - \mathbf{y}_i)^T C (\mathbf{x}_i - \mathbf{y}_i) = 0$$

from which follows that

$$\sum_{i=1}^{k} \frac{(\mathbf{x}_i^*)^T C \mathbf{x}_i^*}{g_i} = \sum_{i=1}^{k} \frac{(\mathbf{x}_i^*)^T C \mathbf{y}_i^*}{g_i} = \sum_{i=1}^{k} \frac{(\mathbf{y}_i^*)^T C \mathbf{y}_i^*}{g_i}$$

and the result follows. $\square$

As the above result holds for any $g \in \Delta_k^*$, we obtain that

$$\text{LOT}_{k,c}(\mu,\mu) = \min_{g \in \Delta_k^*} \min_{\mathbf{x} \in K_{a,g}} \sum_{i=1}^{k} \frac{(\mathbf{x}_i^*)^T C \mathbf{x}_i^*}{g_i}$$

Then by formulating back this problem in term of matrices, we obtain that

$$\text{LOT}_{k,c}(\mu,\mu) = \min_{g \in \Delta_k^*} \min_{Q \in \Pi_{a,g}} \langle C, Q\text{diag}(1/g)Q^T \rangle$$

from which the result follows. $\square$

# C Additional Experiments

## C.1 Comparison of the γ schedules

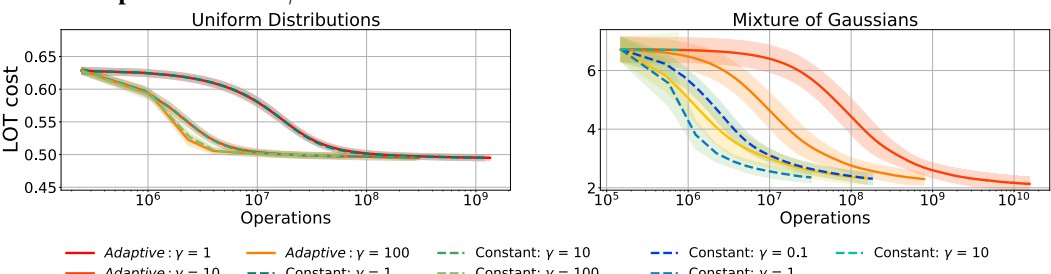

Figure 5: In this experiment, we compare two strategies for the choice of the step-size in the MD scheme proposed by Scetbon et al. [2021] on two different problems. More precisely, we compare the constant γ schedule with the proposed adaptive one and compare them when the distributions are sampled from either uniform distributions (*left*) or mixtures of anisotropic Gaussians (*right*). We show that the range of admissible γ when considering a constant schedule varies from one problem to another. Indeed, in the right plot, we observe that the algorithm converges only when $\gamma \leq 1$, while in the left plot, the algorithm manages to converge for $\gamma \leq 100$. We also observe that our adaptive strategy allows to have a consistent choice of admissible values for γ whatever the problem considered. It is worth noticing that whatever the γ chosen, the algorithm converges towards the same value, however the larger γ is chosen in its admissible range, the faster the algorithm converges.

## C.2 Gradient Flows between two Moons

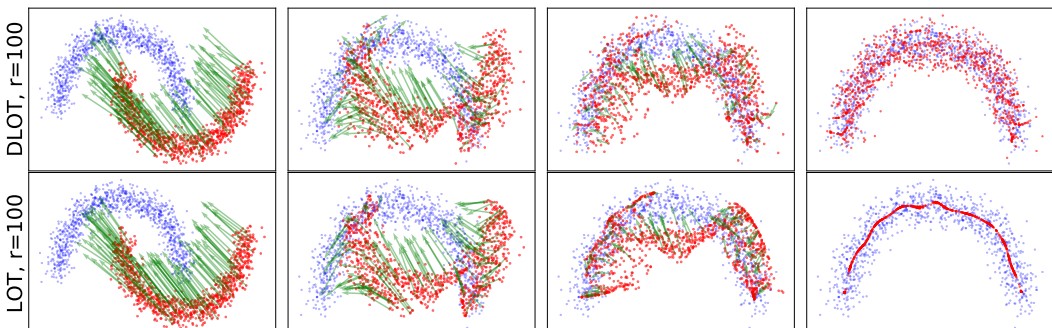

Figure 6: We compare the gradient flows $(\mu_t)_{t \geq 0}$ (in red) starting from a moon shape distribution, $\mu_0$, to another moon shape distribution (in blue), $\nu$, in 2D when minimizing either $L(\mu) \triangleq \mathrm{DLOT}_{r,c}(\mu, \nu)$ or $L(\mu) \triangleq \mathrm{LOT}_{r,c}(\mu, \nu)$. The ground cost is the squared Euclidean distance and we fix $r = 100$. We consider 1000 samples from each distribution and and we plot the evolution of the probability measure obtained along the iterations of a gradient descent scheme. We also display in green the vector field in the descent direction. We show that the debiased version allows to recover the target distribution while $\mathrm{LOT}_{r,c}$ is learning a biased version with a low-rank structure.