# OpenReview forum: "Low-rank Optimal Transport: Approximation, Statistics and Debiasing"
_NeurIPS.cc/2022/Conference — NeurIPS 2022 Accept_

### Official Review · Reviewer_hFtE · 2022-07-02

**Rating:** 7
**Confidence:** 3
**Soundness:** 3 good
**Presentation:** 3 good
**Contribution:** 3 good

**Summary:**

This paper provides an enormous amount of theoretical analysis about low-rank OT: the convergence rate of low-rank OT to the true OT wrt rank parameter, sample complexity for estimating LOT; introduce debiased version of LOT which metrizes the weak convergence; bridge LOT with clustering methods. Practically, they propose a novel initialization to avoid the bad local minima.

**Questions:**

Question:

Figure 2, which stage is the middle two plots of Figure 2 in gradient flow? They seem to be in a very late stage of convergence, but a weird phenomenon is that the gradient flows of both DLOT and LOT exceed the target distribution firstly, and then come back. Especially when looking at those green arrows, they firstly point outside the moon, then point inside the moon. I think if you solve gradient flow correctly, it will not have this "exceed first and then pull back" process.

Typo:

row 142 delete "in"

all the equation (6) is referenced as (16)

**Limitations:**

.

**Strengths And Weaknesses:**

Strengths:

This paper provides the rigorous theoretical analysis of low-rank OT:

- the convergence rate of low-rank OT to the true OT wrt rank parameter,

- sample complexity for estimating LOT is dimensional independent;

- introduce debiased version of LOT which metrizes the weak convergence;

- LOT$(\mu,\mu)$ can be seen as a generalization of the k-means method

- Practically, they also propose a novel initialization to avoid the bad local minima.

Weakness:

- The example in Figure 2 is too simple. Swiss roll or two moons in figure 3.2 is more convincing.

---

> ### Author Response · Authors · 2022-08-02
> **Response from Authors**
>
> > The example in Figure 2 is too simple. Swiss roll or two moons in figure 3.2 is more convincing.
>
> &#8594; We have followed the suggestion made by the reviewer and added an additional experiment in the Supp. Mat. (please refer to sec. C.2, Fig. 6) comparing the GF of LOT and DLOT between two moons. We will add this new experiment in the main text of our final version.
>
> > Figure 2, which stage is the middle two plots of Figure 2 in gradient flow? They seem to be in a very late stage of convergence, but a weird phenomenon is that the gradient flows of both DLOT and LOT exceed the target distribution firstly, and then come back. Especially when looking at those green arrows, they firstly point outside the moon, then point inside the moon. I think if you solve gradient flow correctly, it will not have this "exceed first and then pull back" process.
>
> &#8594; We run a GD scheme during 200 iterations and we plot in the middle the states at 50 and 100 iterations. We will precise it in the final version. We also thank the reviewer for pointing out this observation on the GF: we have corrected our GF by considering a smaller step-size in the GD scheme and we have replaced the figure in our main text (please refer to Fig. 2 of the new submission).
>
> > Typo:
> row 142 delete "in"
> all the equation (6) is referenced as (16)
>
> &#8594; We thank the reviewer for pointing out these typos which have been corrected for the final version.

---

### Official Review · Reviewer_W6vC · 2022-07-09

**Rating:** 7
**Confidence:** 4
**Soundness:** 4 excellent
**Presentation:** 4 excellent
**Contribution:** 3 good

**Summary:**

This work advances the theory of low-rank factorizations for OT by studying the approximation error as a function of rank and the sample complexity of LOT. It additionally proposes the debiased formulation, DLOT, which is shown to interpolate between MMD and OT, and that it metrizes weak convergence. Additional connection to clustering is drawn and better practices of using adaptive stepsizes and better initializations are suggested. Experiments are done to support the claims in 2D synthetic examples as well as on the Newsgroup20 dataset.


**Questions:**

Please refer to my questions in the "Strengths And Weaknesses" section.

**Limitations:**

The author has addressed the limitations and future directions to take to advance LOT. Societal impact is not discussed but I don't think it's needed.

**Strengths And Weaknesses:**

This paper is well-written and easy to follow. Although the contributions are a bit all over the place regarding LOT, they are clearly stated and adequately justified.

On the theory side, the paper provides comprehensive bounds for approximation error and sample complexity while improving the bounds from previous results (e.g. Liu et al. 2021). Then the debiased formulation of LOT is shown to exhibit desirable properties similar to Sinkhorn divergence. The connection to clustering is interesting since it is specific to the low-rank approximation, something that full-rank versions cannot do. While I do not find any of the results surprising or groundbreaking, they are solid and much needed for future research on LOT.

The experiments section is short but verifies part of the theory. There are some missing experiments justifying the adaptive choice of $\gamma_k$ versus without adaptation. Parts of the figures and captions could be improved --- see detailed comments below.

A central question I have regarding the practicality of LOT: Is the computational benefit of LOT worth the introduction of nonconvexity and spurious local minima? I would hope to see more experiments (at least empirically) on demonstrating the benefits gained by low-rank approximation and advice on which $r$ to choose. It seems to me that LOT is only efficient when the ground cost matrix admits a low-rank factorization. In what applications is such condition met?

Comments:
- Line 99: "obtain next a control the approximation", missing "of"?
- Line 126: sample complexity shows promises since it does not depend on dimension - but wouldn't $||c||_\infty$ in Proposition 4 depend in the sense that in many applications the diameter of $\mathcal{X}$ could increase exponentially in $d$? Also as discussed in the paragraph below Proposition 4, $K_r$ could go to infinity.
- Line 187, Line 192: (15) does not exist. Do you mean (6)?
- Line 238: "we do not have access to the this", no "the" here
- Line 252: (15) should be (6)?
- Figure 1: which $r$ is used for the upper bound curve?
- Figure 4: what is the x-axis "operations"? Why do some curves not start at 0 on the x-axis?
- Figure 4: what is the takeaway message from the right figure?

---

> ### Author Response · Authors · 2022-08-02
> **Response from Authors (1/2)**
>
> > The experiments section is short but verifies part of the theory. There are some missing experiments justifying the adaptive choice of γk versus without adaptation.
>
> &#8594; We agree with the reviewer and we have provided an additional experiment in the Supp. Mat. (please refer to sec C.1, Fig. 5) showing the interest of the adaptive step-size in practice. We will add this experiment in the main text of the final version.
>
> > Parts of the figures and captions could be improved --- see detailed comments below.
>
> &#8594; We have followed the suggestions of the reviewer and changed the figures accordingly.
>
> > A central question I have regarding the practicality of LOT: Is the computational benefit of LOT worth the introduction of nonconvexity and spurious local minima?
>
> &#8594; This is indeed the point we have tried to make in this paper. In practice, our experiments suggest (as is often the case for factorized approaches) that only global minima (or at least local minima with a transportation cost very close to the optimal one) are attractive and therefore the non-convexity of the objective does not seem to be an obstacle here. Indeed, in Fig. 4, we show that whatever the initialization considered, the algorithm converges toward the same value. Therefore if we were able to initialize the algorithm close to the global minima we would also converge towards this value, meaning that the value obtained is at least very close to the optimal one. Moreover, experiments in Fig. 1~3 illustrate the above statement as well. In Fig. 1, we observe that our statistic (computed using the algorithm proposed in Scetbon et al. [2021]) converges towards 0 according to the theoretical rates obtained. In Fig. 2, we recover the target distribution meaning that we correctly minimize DLOT (which requires having access to a meaningful gradient of DLOT computed by solving the LOT problems involved in DLOT). Finally, we observe in Fig. 3 (top row) that we recover the same partition as the one obtained by kmeans on various clustering problems.
>
> > I would hope to see more experiments (at least empirically) on demonstrating the benefits gained by low-rank approximation and advice on which r to choose.  It seems to me that LOT is only efficient when the ground cost matrix admits a low-rank factorization. In what applications is such condition met?
>
> &#8594; Our goal here is to bring clearer explanations on the effect of this new regularization on the OT problem and our contributions are mostly theoretical ones. We also want to recall that the goal of such regularization is not to approximate the true OT cost from samples, which is a non-solvable problem in high dimension but rather, as the entropic approach, to obtain a meaningful quantity able to compare distributions in the finite sample regime, even in high dimensions. Indeed recall that when $r=1$, DLOT is exactly the Maximum Mean Discrepancy (which is already a widely used metric in ML) and increasing $r$ allows to capture sharper information about the geometry of the problem instead of considering the “flat” geometry induced by the MMD. The higher the rank is, the more information about the geometry of the problem one gets, yet, at the same time, the more degraded estimation becomes as a result. Therefore, the rank $r$ introduces (much like $\varepsilon$ in entropic OT) a tradeoff, and given a certain precision $\delta$ and a number of samples $n$, the choice of the rank $r$ should be chosen the largest possible such that $\sqrt{r/n}\leq \delta$.
>
> Note that when the data admits a low-rank structure (meaning that the ground cost matrix is low-rank), then it seems empirically that one does not need to choose a rank higher than this intrinsic dimension of the data. This observation deserves more work and we think that it is out of the scope of this paper. In addition, low-rank cost matrices may appears in various setting, especially when data are supported on a low-dimensional manifold with $d \ll n$ where $d$ is the dimension of the manifold and $n$ is the number of samples. A classical illustration of this situation is when the cost considered in the squared Euclidean distance on $\mathbb{R}^d$ for which we have an exact low-rank factorization assuming that $d\ll n$.
>
> > Line 126: sample complexity shows promises since it does not depend on dimension - but wouldn't ||c||∞ in Proposition 4 depend in the sense that in many applications the diameter of X could increase exponentially in d? Also as discussed in the paragraph below Proposition 4, Kr could go to infinity.
>
> &#8594; We agree that the diameter may become larger as we increase the dimension $d$ in some cases. However, our upper bound does not show any dependence in the dimension associated to either the regularization parameter $r$ and most importantly the number of samples $n$.
>
> > Figure 1: which r is used for the upper bound curve?
>
> &#8594; We plot the upper bound using $r=1$. We have precised it in our new submission.

---

> > ### Author Response · Authors · 2022-08-02
> > **Response from Authors (2/2)**
> >
> > > Figure 4: what is the x-axis "operations"?
> >
> > &#8594; The x-axis corresponds to the total number of algebraic operations. This number is computed at each iteration of the outer loop of the algorithm proposed in Scetbon et al., [2021] and is obtained by computing the complexity of all the operations involved in their algorithm to reach it. We consider this notion of time instead of CPU/GPU time as we do not want to be architecture/machine dependent. We have precised it for the final version.
> >
> > > Figure 4: why do some curves not start at 0 on the x-axis?
> >
> > &#8594; Some curves do not start at 0 because we start plotting the curves after obtaining the initial point which in some case requires more algebraic operations (e.g. kmeans methods). We have added an explanation for the final version.
> >
> > > Figure 4: what is the takeaway message from the right figure?
> >
> > &#8594; The right figure of Fig.4 shows two main observations: (i) that the initial point obtained using a “rank 2” or random initialization can be close to spurious and non-attractive local minima, which may trigger the stopping criterion too early and prevent the algorithm from continuing to run in order to converge towards an attractive and well behaved local minimum. (ii) When initialiazing the algorithm using kmeans methods, we show that our stopping criterion is a decreasing function of time meaning that the algorithm converges directly towards the desired solution. We have clarified these two points in our new submission.
> >
> > > Line 99: "obtain next a control the approximation", missing "of"?
> > Line 187, Line 192: (15) does not exist. Do you mean (6)?
> > Line 252: (15) should be (6)?
> > Line 238: "we do not have access to the this", no "the" here
> >
> > &#8594; We thank the reviewer for pointing out these typo which have been corrected for the final version. (Indeed, Eq. (15) in the main text refers to Eq.(6)).

---

> > > ### Comment · Reviewer_W6vC · 2022-08-05
> > > **Response to authors**
> > >
> > > Dear authors,
> > >
> > > Thank you for the very detailed response. This has addressed most of my concerns and I'm raising my score.
> > >
> > > It seems to me there are two important and interesting theoretical questions left unanswered that are suitable for future work: 1) Under what conditions do spurious local minima not exist in LOT formulation? 2) When does the distance matrix of the data have low-rank structure? For 2), if data are indeed supported on a low-dimensional manifold, then should we use the ambient distance or the geodesic distance on the manifold (which is hard to obtain)? If using ambient distance would it still be low-rank?

---

> > > > ### Author Response · Authors · 2022-08-06
> > > > **Thank you for taking the time to read our rebuttal**
> > > >
> > > > Dear Reviewer:
> > > >
> > > > Many thanks for your appreciation and supporting comments. Thank you very much for your suggestions. What follows is a simple and fairly open-ended response on the items you have raised:
> > > >
> > > > On item (1): this is indeed an important subject. One might hope to start from the simplest possible cases in which LOT is shown to provably converge to the global optimum given a fairly simple initializer.
> > > >
> > > > On item (2): if we understand your question correctly, we feel this is a fairly general question on embeddability of general metrics in Euclidean spaces that goes beyond, in our opinion, LOT's study. This does, however, of course play a very important practical role when using LOT beyond the $p=2$ - Wasserstein regime, and is tightly related to item 3:
> > > >
> > > > On item (3): those are extremely important questions indeed for LOT applications beyond the $p=2$ - Wasserstein regime. We have collaborated with people that are interested in using LOT to achieve larger scale computations, and who are tempted to use not only the case $p\ne 2$ but, as you mention, more advanced, composite distance functions. We do not have a clear picture yet of whether these settings are harder to optimize, and, if any difficulty arises, whether it comes from the LOT optimization itself, or from the fact that such distances are harder to approximate with low-rank structures. Such geodesic distances are known to be more robust to outliers (notably when using a NN graph) and so it would be desirable to achieve (and understand) a better performance in such settings.

---

### Official Review · Reviewer_Tiv3 · 2022-07-09

**Rating:** 7
**Confidence:** 4
**Soundness:** 4 excellent
**Presentation:** 2 fair
**Contribution:** 3 good

**Summary:**

The optimal transport (OT) is becoming more and more prominent in machine learning field, however, traditional algorithm such as the linear program has a slow computational speed. In the last decade the entropy-regularized OT (EOT) was proposed and the speed has been  improved a lot. This work studies the low-rank OT (LOT) which is an algorithm proposed by Scetbon $\textit{et al.}$ [2021] that has a promising linear time complexity by searching for the low-cost couplings with low-nonnnegative ranks. The rate of convergence and an dimension independent upper bound of the sample complexity are provided. Furthermore, a debiased version of LOT (DLOT) is proposed, ad the debiasing terms connect LOT to clustering methods. To improve the computation performance adaptive step size and better initializations are introduced, and the effectivenesses are empirically verified by experiments.

**Questions:**

1. The adaptive step size improves the convergence. The authors also suggest clipping the step size in the range of [1, 10] for most use cases. Could an explanation or evidence be provided to support this choice?

2. In Fig. 1, the notations are confusing. Could the authors choose the variable names more carefully so they are consistent with previous sections? Eg. $n$ is number of samples here and in a few places, but somewhere else $n$ represents dimension. Also, is the dimension $d$ the same as stated in line 203 or is it the dimension of the marginal?

3. In Fig. 1, it seems that the DLOT values of larger $r$ are higher among all $d$ cases. Is there an intuition or explanation?

4. Several references of equations are mislabeled. Eg. in line 187, 192, and 252, in the main text the reference equations are (15) while in the supplements (16). Also, there is not eq. (15) in the main text. Please fix and make them consistent.

Minor errors:
[1] Line 141-142: "...one obtained in Proposition in 4...", please remove a redundant "in".
[2] Line 244: A typo: "...we show the iterates obtained by a gradient descent...", should iterates be "iterations"?
[3] Eq. (6): "$Diag$" should be "diag" instead.

**Limitations:**

There is no negative societal impact. The authors address the limitations.

**Strengths And Weaknesses:**

Strengths:
This paper extends the LOT work by Scetbon $\textit{et al.}$ [2021] and studies the theoretical and practical properties of LOT deeply in several aspects. Such complete investigation of an algorithm is essential for bringing in a member into the computational OT family. This work also proposes interesting ideas such as linking the low-rank transport bias to the clustering method, which may inspire other applications and benefit the machine learning community.

Weaknesses:
The naming for the variables and equations referencing labels are confusing, as a result, sometimes it is hard to follow. The clarity of this paper could be improved and the notations could be used in a more consistent way so the readers can understand the meanings of the plots and equations with less efforts. See questions below for more details.

---

> ### Author Response · Authors · 2022-08-02
> **Response from Authors (1/2)**
>
> > The naming for the variables and equations referencing labels are confusing, as a result, sometimes it is hard to follow. The clarity of this paper could be improved and the notations could be used in a more consistent way so the readers can understand the meanings of the plots and equations with less efforts. See questions below for more details.
>
> &#8594; We have followed the suggestions made by the reviewer and clarified the notations as well as the referencing of the equations used in the paper.
>
> > The adaptive step size improves the convergence. The authors also suggest clipping the step size in the range of [1, 10] for most use cases. Could an explanation or evidence be provided to support this choice?
>
> &#8594; The main issue when choosing a constant step size is that the range of admissible $\gamma$ such that the algorithm converges depends on the problem considered. Indeed, for some problems we have encountered in practice, the algorithm might fail to converge for large $\gamma$. This is because the algorithm of Scetbon et al. [2021, Alg. 3] requires solving Eq. (7) at each iteration, which involves some kernels (defined in l.198, 199, and 200. of the old version). These kernels depend on both  $\gamma$ and the current couplings. If their product (e.g. $\gamma CR_k diag(1/g_k)$) have large values, then taking $\exp(-\gamma CR_k diag(1/g_k))$ will result in a kernel with some zero entries (as can be often the case for the Sinkhorn algorithm with low regularization). This is a real issue when solving (7) using the Dykstra’s algorithm proposed in Scetbon et al. [2021, Alg. 2], which divides (as the Sinkhorn algorithm does) quantities by these kernels: if one of these kernels have $\sim 0$ entries, this may result in a “divison by 0” overflow error. Therefore $\gamma$ must be chosen such that at each iteration $\gamma CR_k diag(1/g_k)$ has reasonable entries.
>
> In order to alleviate this issue and obtain a generic range of admissible values for $\gamma$ independently of the problem considered, we propose to use an adaptive step-size. By doing so, we are able to guarantee a lower-bound of the exponential term involved in the expression of the kernels at each iteration. Indeed, recall that our adaptive step-size is defined at each iteration as follows (Eq. (8) in the paper):
> $$\gamma_k = \gamma / \Vert (CR_kdiag(1/g_k),C^TQ_k diag(1/g_k), -\omega_k / g_k^2  \Vert_{\infty}$$
>
> where $\gamma$ is constant along the iterations. Then at each iteration we can guarantee that:
> $$ 0 \leq \Vert \gamma_k CR_k diag(1/g_k)\Vert_{\infty}, \Vert \gamma_k C^TQ_k diag(1/g_k)\Vert_{\infty},\Vert \gamma_k \omega_k / g_k^2 \Vert_{\infty} \leq \gamma$$
>
> meaning that coordinatewise we obtain
>
> $$ \exp(-\gamma_k * CR_k diag(1/g_k)), \exp(- \gamma_k * C^TQ_k diag(1/g_k)), \exp(\gamma_k \omega_k / g_k^2) \geq \exp(-\gamma)\; .$$
>
> By fixing the range of $\gamma$ to be $[1,10]$, we can now guarantee that whatever the problem considered, the exponential terms involved in the kernels do not admit values smaller than \exp(-10).
> We observe empirically that it is sufficient in order to perform all the operations of the Dykstra’s algorithm solving Eq. (7) and to obtain convergence. We have clarified this point in our new submission.
>
>
> We also have added a new experiment in the Supp. Mat. (please refer to sec. C.1, Fig.5) demonstrating the interest of using such an adaptive step-size and showing that the range of admissible $\gamma$ may vary from one problem to another if we consider a fixed $\gamma$ schedule. We will add it in the main text for the final version.
> Note also that the convergence of the MD scheme using such adaptive step-size is also theoretically justified (see results of~D’Orazio et al. [2021] and Bayandina et al. [2018]).
>
> > In Fig. 1, the notations are confusing. Could the authors choose the variable names more carefully so they are consistent with previous sections? Eg. n is number of samples here and in a few places, but somewhere else n represents dimension. Also, is the dimension d the same as stated in line 203 or is it the dimension of the marginal?
>
> &#8594; We have followed the suggestions made by the reviewer. More precisely, we always reference $n$ as being the number of samples and $d$ the dimension of the space where are supported the measures. We also have changed the notation of the low-rank associated to the cost matrix (l.205 of the new submission) and we have denoted it: $q$. Note that for the squared Euclidean distance, $q=d+2$.
>
> > In Fig. 1, it seems that the DLOT values of larger r are higher among all dcases. Is there an intuition or explanation?
>
> &#8594; Indeed, this observation was expected according to the rates obtained. We show that the rates should scale in $\sqrt{r/n}$, therefore the higher the rank, the slower it should converge. We have precised it in our new submission.

---

> > ### Author Response · Authors · 2022-08-02
> > **Response from Authors (2/2)**
> >
> > > Several references of equations are mislabeled. Eg. in line 187, 192, and 252, in the main text the reference equations are (15) while in the supplements (16). Also, there is not eq. (15) in the main text. Please fix and make them consistent.
> >
> > &#8594; We thank the reviewer for pointing out these annotation errors. Line 187, 192, and 252 refer to Eq.(6) and we have corrected them in our new submission.
> >
> > > [1] Line 141-142: "...one obtained in Proposition in 4...", please remove a redundant "in".
> > [2] Line 244: A typo: "...we show the iterates obtained by a gradient descent...", should iterates be "iterations"? [3] Eq. (6): "Diag" should be "diag" instead.
> >
> > &#8594; We thank the reviewer for revealing these typos that we have corrected in our new submission.

---

### Official Review · Reviewer_PXvB · 2022-07-16

**Rating:** 7
**Confidence:** 4
**Soundness:** 3 good
**Presentation:** 3 good
**Contribution:** 3 good

**Summary:**

This papers is concerned with a model that approximates optimal transport (OT) using the low-rank coupling/matrices. This model in itself has been proposed a couple of years ago and this paper aims at answering important theoretical questions such as approximation error with respect to standard OT and statistical rates of estimation. They introduce a "debiased" version of their estimation, similar to the one proposed for entropic OT and make the link with clustering methods. Their theoretical work is also complemented with additional tricks for improving the numerical efficiency of the method.


**Questions:**

My main concern is about the results for statistical estimation.

Propositions 4 and 5 is about a one-sided inequality, i.e. there is no absolute value in the left-hand side of Equation in Prop. 4, neither in Equation in Prop. 5. Note that a complete result on statistical estimation is really about both lower and upper bounds. However, the authors only give an upper bound, as written line 123.
Although I think it is true, I do not see how to get a lower bound. It is likely I may have missed a result in the paper showing that it is a simple consequence, and it was maybe obvious for the authors, and it would be meaningful to include it.

Can the authors clarify their result?

As a side remark, it is rather borderline practice that the authors pretend to have a complete result on statistical complexity.
Indeed they write after proposition 4: "This result shows that the estimation of LOTr,c is independent of the dimension and can be performed on general compact metric spaces."
However the result in its current form is only partial and thus one cannot claim anything on statistical estimation. Indeed, the fluctuations of the opposite quantity may be much larger and dependent on the dimension. This can happen in practice.

So what am I misunderstanding here?

Others:
- Is it possible to add the following result. If $\mu_n \to \mu$ for the weak-* topology then $LOT(\mu_n) \to LOT(\mu)$?
- proof of proposition 1: the decomposition of pi line 423 in supplementary (btw, there is a typo there) should be explained a bit more. Is it a standard SVD?
- typos in the supplementary material can be corrected, line 480, line 494.


**Limitations:**

no particular comments.

**Strengths And Weaknesses:**

Although one can argue about the usefulness of the model studied by the authors, it is a good paper by its many theoretical contributions exploring the foundations of low-rank approximation of OT. Results are ranging from obvious, easy to non-trivial and the paper will be a reference for other research developments around this model.
Strengths:
- Paper is well written.
- Several meaningful theoretical results.
- So far I checked, the proofs are correct (I did not check proof of proposition 5).

Weakness:
- See my question below: Authors must address my question below, if not I'll revise my rating accordingly.

---

> ### Author Response · Authors · 2022-08-02
> **Response from Authors**
>
> > Propositions 4 and 5 is about a one-sided inequality, i.e. there is no absolute value in the left-hand side of Equation in Prop. 4, neither in Equation in Prop. 5. Note that a complete result on statistical estimation is really about both lower and upper bounds. However, the authors only give an upper bound, as written line 123. Although I think it is true, I do not see how to get a lower bound. It is likely I may have missed a result in the paper showing that it is a simple consequence, and it was maybe obvious for the authors, and it would be meaningful to include it. Can the authors clarify their result?
>
> &#8594; You are absolutely right, we did not manage to lower bound the plug-in estimator using the true LOT. This result requires additional work and for now,  we do not see how to achieve it. Our proof relies on a specific construction of an admissible and random coupling between the empirical distributions (introduced l.459 of the old version) for which its cost converges at a rate $\mathcal{O}(\sqrt{r/n})$ to the true LOT. Indeed we obtain a control of the difference in absolute value between the true LOT cost and the cost associated to this random coupling. The main issue is that we did not manage to control the error between the cost associated to this coupling and the LOT cost between the empirical measures. Therefore we exploit optimality in order to control at least one side of the difference. Using such technique, we are able to provide an upper-bound of the plug-in estimator of LOT which converges at a parametric rate towards the true LOT independently of the dimension. To the best of our knowledge, it is the first time that such a control has been obtained. The closest current result presented in the literature concerns the statistical control of the transportation cost between a fixed and arbitrary measure with a support of size $r$ and the empirical measure associated to a target probability measure for the specific case of the quadratic cost [Forrow et al., 2019, Theorem 4].
>
> > As a side remark, it is rather borderline practice that the authors pretend to have a complete result on statistical complexity. Indeed they write after proposition 4: "This result shows that the estimation of LOTr,c is independent of the dimension and can be performed on general compact metric spaces." However the result in its current form is only partial and thus one cannot claim anything on statistical estimation.
>
> &#8594; Although we are upfront about the fact (e.g. l.123) that we obtain an upper bound, we understand the concern of the reviewer, and we have clarified this further in our new submission. More precisely, we have replaced the sentence:\
> “This result shows that the estimation of $LOT_{r,c}$ is independent of the dimension and can be performed on general compact metric spaces.“
>
> by
>
> "This result is, to the best of our knowledge, the first attempt at providing a statistical control of low-rank optimal transport. We provide an upper-bound of the plug-in estimator which converges towards $LOT_{r,c}$ at a parametric rate and which is independent of the dimension on general compact metric spaces. While we fall short of providing a lower bound that could match that upper bound, and therefore provide a complete statistical complexity result, we believe this result might provide a first explanation on why, in practice, $LOT_{r,c}$ displays experimentally better statistical properties than unregularized OT and its curse of dimensionality [Dudley, 1969] etc.."
>
> In addition, in order to avoid any confusion, we have clearly stated in the new submission our results presented in Proposition 4 and 5 as follows:
> $LOT_{r,c}(\hat{\mu},\hat{\nu}) \leq LOT_{r,c}(\mu,\nu) + rate$.
>
> > Is it possible to add the following result. If $\mu_n\rightarrow \mu$ for the weak-* topology then $LOT(\mu_n)\rightarrow LOT(\mu)$?
>
> &#8594; In fact, we show in a Proposition presented in the Supp. Mat. (l.550 of the old version), the result mentioned. We have moved it into the main text of our new submission.
>
> > proof of proposition 1: the decomposition of pi line 423 in supplementary (btw, there is a typo there) should be explained a bit more. Is it a standard SVD?
>
> &#8594; Concerning the proof of Prop.1, we have added more explanation in order to make it clearer. In fact, it is not the SVD as we require that $(q_i, r_i)_{I=1}^n$ are nonnegative and sum to 1. We obtain such factorization by simply saying that the nonnegative rank of a nonnegative matrix of size $n\times m$ cannot exceed $\min(n,m)$.
>
> > typos in the supplementary material can be corrected, line 480, line 494.
>
> &#8594; We thank the reviewer for pointing out these typos which we have corrected for the final version.

---

> > ### Comment · Reviewer_PXvB · 2022-08-05
> > **Is statistical complexity of LOT still an open question?**
> >
> > I thank the authors for their explanations. I do think the upper-bound on the statistical complexity is a nice contribution, as well as the rest of the paper. Still, I think a clarification is needed.
> >
> > It is sometimes possible to design estimators that over-estimate or under-estimate quantities of interest. It can be the case that the best lower-bound that can be found is cursed with the dimension, like $1/n^{1/d}$ (although I tend to think that a parametric rate is more likely to occur). As a consequence, there is no concrete result on statistical complexity of LOT in this paper due to this lack of lower bound.
> >
> > The way it is phrased in the paper is very unclear, actually not at all present. What do you think?

---

> > > ### Author Response · Authors · 2022-08-06
> > > **We agree with your assessment.**
> > >
> > > We thank the reviewer for having taken the time to read our rebuttal, and for your reply.
> > >
> > > We do agree with your original assessment, in your first review, and the more recent one above. This is why, as mentioned above, we have replaced the original sentence in the paper:
> > >
> > > *“This result shows that the estimation of $LOT_{r,c}$ is independent of the dimension and can be performed on general compact metric spaces.“*
> > >
> > > by
> > >
> > > *"This result is, to the best of our knowledge, the first attempt at providing a statistical control of low-rank optimal transport. We provide an upper-bound of the plug-in estimator which converges towards $LOT_{r,c}$ at a parametric rate and which is independent of the dimension on general compact metric spaces. While we fall short of providing a lower bound that could match that upper bound, and therefore provide a complete statistical complexity result, we believe this result might provide a first explanation on why, in practice, $LOT_{r,c}$ displays experimentally better statistical properties than unregularized OT and its curse of dimensionality [Dudley, 1969] etc.."*
> > >
> > > We do feel this is a significantly more nuanced formulation, which, while underlining that this is a first result towards a better statistical understanding, mentions that only half the job is done. Is there anything in that formulation that you feel is ambiguous?
> > >
> > > If no, you mentioned **The way it (statistical complexity) is phrased in the paper is very unclear, actually not at all present.**.
> > >
> > > While we have corrected the part mentioned above, it is very likely, given the short time span of the rebuttal, that some other parts of the paper use the former, more forceful formulation (notably abstract or statement of contributions that we have not corrected yet). If you are referring to such parts to make the call above, rest assured that we will correct *all* references to statistical complexity, and introduce the nuance above, which reflects our current thinking.

---

### Author Response · Authors · 2022-08-02
**Response from Authors to Committee and Reviewers**

We thank the reviewers for their thorough reading of our work. We have used their remarks to improve our draft. Please refer to the new submission where the modifications have been marked in blue. We also respond to each of the reviewers below.

---

### Meta-Review · Area_Chair_H2xi · 2022-08-23

**Recommendation:** Accept
**Confidence:** Certain

**Metareview:**

Overall: The paper focuses on advancing our knowledge, understanding and practical ability to leverage low-rank factorizations in optimal transport.

Reviews: The paper received four reviews. 4 accepts (all confident). It seems that there are several reviewers that will champion the paper for publication. The reviewers found the paper is clear and has a clean presentation. The findings are interesting. The authors have provided extensive answers to reviewers' comments, answering most of them successfully.

After rebuttal: A subset of the reviewers engaged in a consensus that the paper should be accepted.

Confidence of reviews: Overall, the reviewers are confident. We will put more weight to the reviews that got engaged in the rebuttal discussion period.

**Award:**

No

---

### Decision · Program_Chairs · 2022-09-14

Accept